# Physiology and ecology combine to determine host and vector importance for Ross River virus

Morgan P Kain[1,2†]*, Eloise B Skinner[1,3†]*, Andrew F van den Hurk[4], Hamish McCallum[3], Erin A Mordecai[1]

[1]Department of Biology, Stanford University, Stanford, United States; [2]Natural Capital Project, Woods Institute for the Environment, Stanford University, Stanford, United States; [3]Centre for Planetary Health and Food Security, Griffith University, Gold Coast, Australia; [4]Public Health Virology, Forensic and Scientific Services, Department of Health, Brisbane, Australia

**Abstract** Identifying the key vector and host species that drive the transmission of zoonotic pathogens is notoriously difficult but critical for disease control. We present a nested approach for quantifying the importance of host and vectors that integrates species' physiological competence with their ecological traits. We apply this framework to a medically important arbovirus, Ross River virus (RRV), in Brisbane, Australia. We find that vertebrate hosts with high physiological competence are not the most important for community transmission; interactions between hosts and vectors largely underpin the importance of host species. For vectors, physiological competence is highly important. Our results identify primary and secondary vectors of RRV and suggest two potential transmission cycles in Brisbane: an enzootic cycle involving birds and an urban cycle involving humans. The framework accounts for uncertainty from each fitted statistical model in estimates of species' contributions to transmission and has has direct application to other zoonotic pathogens.

**\*For correspondence:**
kainm@stanford.edu (MPK);
ebskinn@stanford.edu (EBS)

[†]These authors contributed equally to this work

**Competing interests:** The authors declare that no competing interests exist.

## Introduction

More than 60% of existing infectious diseases of humans are multi-host pathogens (i.e. moving between non-human and human populations) and approximately 75% of emerging and re-emerging infectious diseases affecting humans have a non-human origin (*Taylor et al., 2001*; *van Doorn, 2014*). It it therefore critical to identify the role that different vertebrate host and vector species play in maintaining transmission and facilitating spillover into humans. However, identifying which species enable pathogen persistence and quantifying the relative contribution that each species makes to transmission is notoriously difficult, particularly because definitions for vectors and hosts vary greatly within the literature (*Appendix 1—table 1*). The dynamics of multi-host pathogen systems can range in complexity from spillover between a single source population to a single target population (e.g. from bats to humans as has been postulated for SARS-CoV-1 and SARS-CoV-2: *Boni et al., 2020*) to large interconnected networks of species that maintain a pathogen in a given environment and facilitate spillover into a target population (e.g. zoonotic arboviruses, such as West Nile (WNV) and Rift Valley fever (RVFV) viruses: *Viana et al., 2014*).

Developing appropriate mitigation strategies for zoonotic pathogens hinges on quantifying which processes have the largest influence over each species' importance in transmission cycles. Studies characterising zoonotic arbovirus transmission often focus on pairwise transmission between non-human hosts and vectors, or vectors and humans (for example work in WNV: *Marm Kilpatrick et al., 2006*, Ross River virus: *Koolhof and Carver, 2017*, *Stephenson et al., 2018*, leishmaniasis:

*Stephens et al., 2016*, Chagas disease: *Gürtler and Cardinal, 2015*, *Jansen et al., 2018*). However, these and other proposed approaches (*Appendix 1—table 1*) that capture only a portion of a pathogen's transmission cycle cannot completely quantify a species' contribution to transmission within a community. Understanding the ecological importance of host and vector species for transmission requires modeling the complete transmission cycle (host-vector-host or vector-host-vector transmission), 'closing the loop' by estimating the number of new infections in the next generation. This is needed to quantify each species' contribution to $\mathcal{R}_0$, defined as the number of new infections arising from a single case in an otherwise susceptible population. While this is well understood (e.g. see *Turner et al., 2013*, *Fenton et al., 2015*, *Webster et al., 2017*), this approach is used less frequently for multi-vector, multi-host pathogens because of the need for data across multiple phases of transmission for multiple host and vector species.

Here, we present a general framework (*Box 1*) that: (1) quantifies host and vector species' relative importance across a complete transmission cycle of zoonotic arboviruses (*Figure 1*), using Ross River virus (RRV) as the model virus—a system for which we have data for many host and vector species for nearly all components of the transmission process; (2) identifies which of the many interacting physiological and ecological processes have the largest control over the importance of each species; and (3) helps to reveal where the largest sources of uncertainty occur in order to identify which datasets require additional collection for more robust predictions (*Restif et al., 2012*). The approach uses three nested metrics of increasing biological complexity: physiological competence; transmission over one half of the pathogen's life cycle (half-cycle transmission; that is, host-to-vector or vector-to-host transmission); and transmission over the pathogen's complete life cycle (complete-cycle transmission) (*Box 1*). This strategy has application to other zoonotic pathogens for which some physiological and ecological data exist across vectors and hosts. Even for systems with limited data, a framework that integrates the entire transmission cycle can be useful for hypothesis testing and for guiding data collection by identifying the processes that most contribute to uncertainty in competence (i.e. model-guided fieldwork, sensu *Restif et al., 2012*).

As a case study, we focus on RRV, an alphavirus that causes a disease syndrome characterized by polyarthritis, which is responsible for the greatest number of mosquito-borne human disease notifications in Australia, with approximately 5000 cases notified annually (*Australian Govt. Dept. of Health, 2020*). It has also caused major epidemics in Pacific Islands involving tens of thousands of cases (*Aaskov et al., 1981*; *Tesh et al., 1981*; *Harley et al., 2001*), and may have the potential to emerge and cause explosive epidemics out of its current geographical range (*Flies et al., 2018*; *Shanks, 2019*). Understanding the drivers of epidemic and endemic transmission of RRV in Australia and Pacific Island countries has remained challenging because of the number of hosts and mosquitoes that potentially become infected and the large uncertainty around which of these vectors and hosts contribute most to transmission. Under controlled laboratory conditions, more than 15 species of mosquitoes from at least five genera have demonstrated the physiological ability to transmit RRV. The disease has long been considered to exist in a zoonotic transmission cycle, primarily because the number of human cases during winter months was considered to be too low to sustain community transmission (*Harley et al., 2001*). However, the most important vertebrate hosts of RRV are highly ambiguous because more than 50 species have demonstrated serological evidence of natural exposure to RRV (reviewed in *Stephenson et al., 2018*). Much uncertainty remains as to which vertebrate species contribute to RRV community transmission and how the importance of these species in transmission varies by locations (such as urban vs. rural settings, or in Australia vs. the Pacific Islands, where there are different vertebrate communities). Although insights have previously been gained through modeling approaches (*Carver et al., 2009*; *Denholm et al., 2017*; *Koolhof and Carver, 2017*), these studies note that future progress in RRV modeling requires consideration of the dynamics of multiple mosquito species and multiple hosts, accounting for their differing availability and physiological capability to transmit RRV.

We parameterize our framework for RRV to quantify the relative importance of hosts and vectors for disease transmission and to illustrate how the relative importance of these species changes depending on what metric is used. Specifically, we ask the following questions for RRV transmission in Brisbane, Australia, a community in which RRV is endemic:

1. Which host and vector species are most physiologically competent for transmitting RRV?

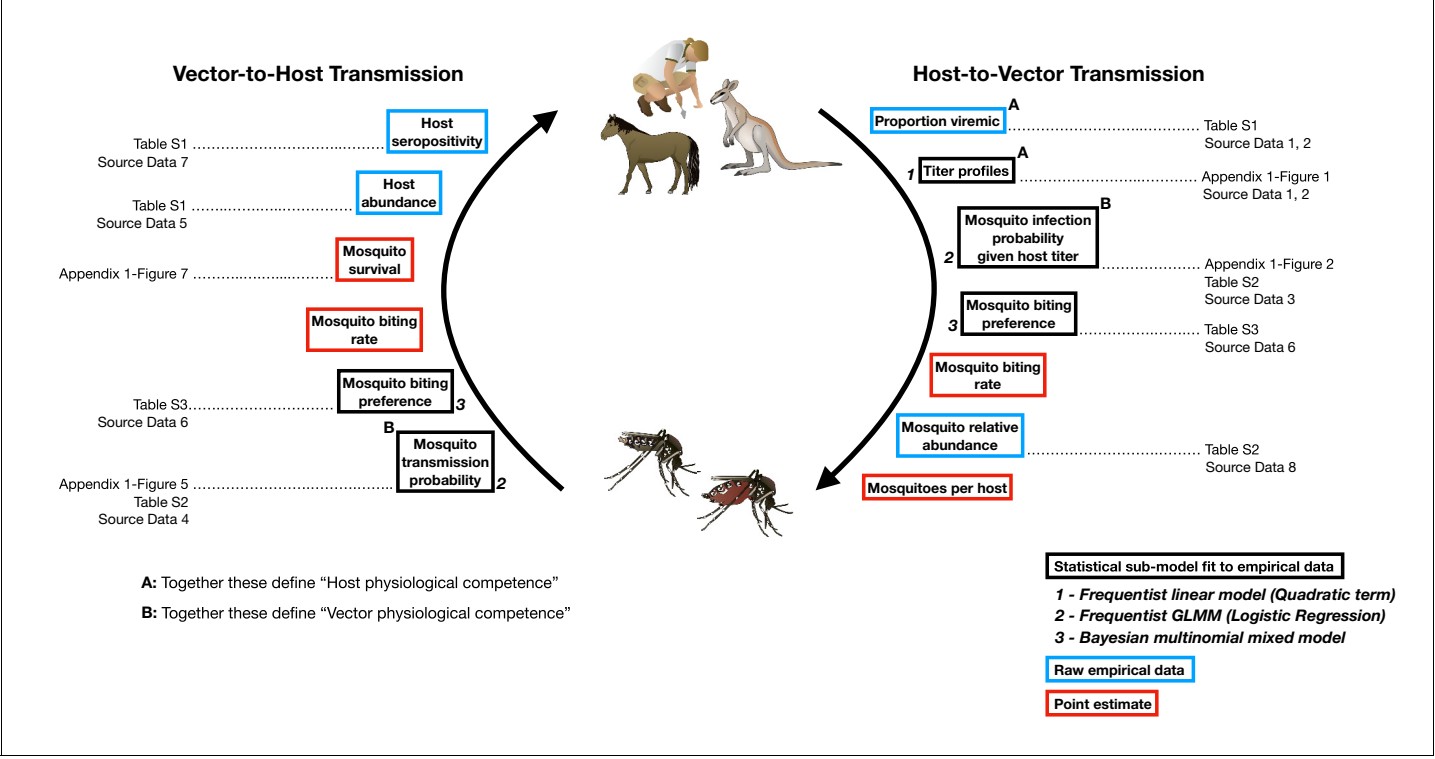

**Figure 1.** The transmission cycle of Ross River virus, a multi-host, multi-vector arbovirus, and the components our framework uses to model this transmission cycle. The first requirements for transmission are physiologically competent hosts that become infected (**A**: 'proportion viremic') and are able to replicate the virus to suitable levels to infect vectors (**A**: 'titer profiles') and vector species that can become infected (**B**: 'Mosquito infection probability') and eventually are able to transmit virus (**B**: 'Mosquito transmission probability'). Physiologically competent hosts and vectors contribute to the transmission of the virus through a continuous cycle of transmission, which can be viewed from two perspectives, either starting with an infected host or starting with an infected vector; regardless of perspective, a single complete cycle contains a single set of physiological and ecological components. Each of these components are used in our framework in one of three ways: statistical models fit to empirical data, from which uncertainty is propagated into the final calculations of transmission (boxes outlined in black); raw empirical data (boxes outlined in blue); and point estimates (boxes outlined in red). Italic bold numbers and text next to the boxes outlined in black describe, in brief, the type of statistical model used to estimate each component (GLMM stands for generalized linear mixed model). Details on all components are provided in the Materials and Methods, Supplementary files, and Appendix Figures that are listed next to framework components; associated raw Source Data files are also listed.

2. How does integrating species ecology change the most important hosts and vectors when considering a half (host-to-vector or vector-to-host) or complete (host-vector-host or vector-host-vector) transmission cycle?
3. How do viruses circulate through different species in the community, that is, which hosts and vectors contribute to intra- and inter-species transmission?

## Results

### Physiological competence
#### Host competence
To quantify a host species' physiological competence we multiplied the proportion of individuals of that species that developed a viremic response by the area under that species' estimated titer profile over time, which we fit to the individuals that mounted a viremic response. This AUC metric captures both the absolute magnitude and duration of a host species' viremic response, weighted by how common this response is. Of the vertebrate species available for the analysis in Brisbane, we estimated that rats and macropods had the strongest viremic response to RRV infection (*Figure 2A*). Sheep, rabbits, humans, and possums formed a distinct cluster of hosts with the next strongest responses; uncertainty in host titer profiles obscures our ability to differentiate among the responses

**Table 1.** Model components, the transmission metrics in which they are used, and the data and statistical modeling choices used to estimate each.

The column 'Parameter' lists the parameters as they appear in *Equation 1* and *Equation 2*. Abbreviations for the transmission metrics are: HC = host competence; H-to-V = host-to-vector transmission; V-to-H = vector-to-host; H-to-H = host-vector-host; V-to-V = vector-host-vector. The 'Data' column lists the name of the Source data file containing the raw data; all citations are listed in the online supplement (*Supplementary file 3*). Data sources are described in the Supplemental Methods: Data. The 'Methodological Details' column lists where in the manuscript methods are described.

| Model Component | Parameter | Transmission Metrics | Data | Statistical Model | Uncertainty | Methodological Details |
|---|---|---|---|---|---|---|
| Proportion of individuals of host species *i* exposed to infection that produce viremia | $\omega_i$ | HC H-to-V H-to-H V-to-V | host_response. csv human_titer.csv | Raw Data | None (Raw Data) | *Methods: Vertebrate hosts: titer profiles; Supplemental Methods: Host physiological competence;* **Supplementary file 1** |
| Host titer (in species *i* on day *j*) | $\theta_{id_i}$ | HC H-to-V H-to-H V-to-V | host_response. csv human_titer.csv | Linear model with a quadratic term for days post infection | 1000 simulated titer curves for each species | *Methods: Vertebrate hosts: titer profiles; Supplemental Methods: Host physiological competence;* **Appendix 1—figure 1;** **Supplementary file 1** |
| Proportion of host species *i* that are seronegative | $\eta_j$ | V-to-H H-to-H V-to-V | host_ seroprevalence. csv | Raw Data | None (Raw Data) | **Supplementary file 1** |
| Infection probability of mosquito species *j* as a function of dose | $p_j$ | VC H-to-V V-to-H H-to-H V-to-V | mosquito_ infection.csv | Generalized linear model (logistic regression) | 1000 samples from a multivariate Normal distribution using the estimated means and vcov matrix | *Mosquito vectors: infection and transmission probability; Supplemental Methods: Vector physiological competence;* **Appendix 1—figure 3;** **Supplementary file 2** |
| Transmission probability of mosquito species *j* *r* days post infection | $p_{ir_j}$ | VC V-to-H H-to-H V-to-V | mosquito_ transmission. csv | Generalized linear model (logistic regression) | 1000 samples from a multivariate Normal distribution using the estimated means and vcov matrix | *Mosquito vectors: infection and transmission probability; Supplemental Methods: Vector physiological competence;* **Appendix 1—figure 4;** **Supplementary file 2** |
| Survival probability of mosquito species *j* up to *r* days post infection | $\lambda_{jr_j}$ | V-to-H H-to-H V-to-V | – | Exponential decay using point estimate for daily mortality probability | None | *Methods: Mosquito survival;* **Appendix 1—figure 7** |
| Proportion of mosquito species *j*'s blood meals that are obtained from host species *i* | $\dfrac{\beta_{ij}\alpha_i}{\sum_{i=1}^{I}\beta_{ij}\alpha_i}$ | V-to-H H-to-H V-to-V | mosquito_ feeding.csv host_ abundance.csv | Custom Bayesian regression model | Bayesian posterior | *Methods: Mosquito feeding preference; Supplemental Methods: Mosquito feeding preference;* **Supplementary file 2;** **Supplementary file 3** |
| Number of susceptible mosquitoes of species *i* per host species *j* | $\phi_{ij}$ | H-to-V H-to-H V-to-V | mosquito_ abundance.csv | Raw Data + Assumption | None (Raw Data + Point Estimate) | |
| Daily biting rate of mosquito species *j* | $\sigma_j$ | H-to-V V-to-H H-to-H V-to-V | – | Assumption | None (Point Estimate) | Assumed value of 0.5 Day$^{-1}$ |

of these species. Of the remaining species, we estimated that 'birds' (an average of *Gallus gallus domesticus* [Chicken], *Cacatua sanguinea* [Little corella], and *Anas superciliosa* [Pacific black duck]) had a stronger viremic response than flying foxes, horses, and cattle. No dogs or cats developed detectable viremia when exposed to RRV experimentally (N = 10 for each species), resulting in the lowest physiological competence. Fitted titer profiles for all hosts for which data were available are presented in *Appendix 1—figure 1* (AUC for these profiles are presented in *Appendix 1—figure 2*), while the proportion of the cohort of each host species that developed a viremic response when exposed to RRV is listed in *Supplementary file 1*.

## Box 1. Nested approach for characterising the complete transmission cycle of zoonotic arboviruses.

Stage 1: Physiological competence.
Characterizing the physiological response a species has to infection is fundamental to estimating its potential as a host or vector within a community. We define the physiological competence of a host species as its viremic response to infection multiplied by the proportion of individuals of that species that develop a viremic response when exposed to infection. We model each host species' viremic response as a continuous function over time (*Appendix 1—figure 1*); to compare hosts' physiological competences, we summarize their titer profiles using the area under the curve (AUC), which simultaneously captures the magnitude and duration of titer (*Appendix 1—figure 2*). For vectors, we quantify physiological competence using the product of the proportion of individuals that get infected following exposure to a given dose (*Appendix 1—figure 3*) and the proportion that go on to transmit the virus (*Appendix 1—figure 4*). Specifically, we quantify physiological vector competence using the multiplication of the AUC of these two curves (*Appendix 1—figure 5*, *Appendix 1—figure 6*). For a visualization of these components within an arbovirus life cycle see *Figure 1*.

Stage 2: Transmission over one half of the pathogen's life cycle (host-to-vector or vector-to-host transmission).
To begin to understand the role species play in community transmission, we quantify how many vectors an infected host will generate or how many new host infections an infected vector will create. To do this, we combine host and vector physiological competence (*Stage 1*) with host and vector abundances and contact rates. Specifically, to quantify host-to-vector transmission we combine estimates (while propagating uncertainty) from host titer profiles over time, mosquito infection probabilities given titer (infectious dose), mosquito feeding behavior (which combines vector preference and host abundance), and mosquito abundance (*Figure 1*). For vector-to-host transmission, we combine estimates from mosquito transmission probabilities, survival, mosquito feeding behavior, and host abundance.

Stage 3: Transmission over the pathogen's complete life cycle (host-vector-host or vector-host-vector transmission).
A complete transmission cycle can be achieved by multiplying the two half-transmission calculations from *Stage 2* (host-to-vector and vector-to-host) in either order; the $\mathcal{R}_0$ calculated from either order will be identical. However, each of the two multiplication orders reveals something different. Multiplying host-to-vector by vector-to-host transmission gives host-vector-host transmission (a complete transmission cycle from the perspective of a host), which can be used to reveal all host-to-host pairwise transmission pathways. In other words, beginning with an infected host, how many (and which) other hosts become infected? Conversely, multiplying vector-to-host transmission by host-to-vector will reveal all vector-to-vector transmission pathways starting with an infected vector.

### Vector competence

To quantify mosquito physiological competence, we used the area under the infection probability versus dose curve multiplied by the area under the transmission probability over time since infection curve. We estimated that the mosquito species with the highest physiological potential for RRV transmission (susceptibility of mosquitoes to infection, and of those that become infected, their potential to transmit RRV) was *Coquillettidia linealis*, although the 95% CI for this species overlaps with four species with the next highest median estimates (*Aedes procax*, *Verrallina funerea*, *Aedes vigilax*, and *Mansonia uniformis*) (*Figure 3A*). In contrast, *Culex annulirostris*, *Culex quinquefasciatus*, *Aedes notoscriptus*, and *Culex sitiens* were estimated to all have low physiological potential. Infection probability curves for all mosquito species for which we gathered data, including those in the Brisbane community and from elsewhere in Australia, are shown in *Appendix 1—figure 3* and *Appendix 1—figure 5*.

## Half-transmission cycle

### Host-to-vector transmission

Integrating host physiological competence with ecological factors governing host-vector contacts (see *Figure 1* and *Box 1*) can dramatically change estimated host importance (*Figure 2B*). Despite large uncertainty in estimates for the number of mosquitoes that a single infected host can infect while infectious, humans have both the largest estimated median and highest estimated potential (upper 95% CI bound) for infecting mosquitoes in Brisbane. We predict that an infected human would predominantly infect *Ae. vigilax*, followed by *Ae. procax* and *Cx. annulirostris*. Both rats and macropods, which had the highest physiological potential for transmission (*Figure 2A*), dropped beneath possums, birds, and horses according to median estimates, though overlapping 95% CIs obscure our ability to determine which host is able to infect more mosquitoes while infectious. Similarly, sheep dropped from being in the cluster of the most important species when using physiological response alone (*Figure 2A*) to one of the lowest potential hosts for RRV transmission to mosquitoes in Brisbane (*Figure 2B*). Conversely, horses, which had one of the lowest estimated viremic responses, increased in importance when considering the contribution of ecological traits to community transmission. Cats and dogs were estimated to be unable to transmit RRV to any mosquitoes given that neither mount a viremic response.

### Vector-to-host transmission

While host relative importance markedly changed between physiological competence and transmission over half a transmission cycle, mosquito estimates did not. *Cq. linealis*, *Ae. procax*, *Ae. vigilax*, and *Ve. funerea* were estimated to infect the largest number of hosts (using median estimates) after embedding mosquito physiological competence into vector-to-host transmission (*Figure 3B*), although wide overlapping 95% CI make it impossible to differentiate among these species. We estimated that an infected *Cq. linealis* would mostly infect birds, while an infected *Ae. procax* and *Ae. vigilax* would infect a larger diversity of host species including birds, humans, and dogs. Of the remaining species, *Cx. annulirostris*, *Cx. quinquefasciatus*, and *Cx. sitiens* remained poor vectors, infecting only a small number of hosts.

## Complete-transmission cycle

We calculated the number of second generation hosts an infected host would infect (or the number of second generation mosquitoes an infected mosquito would infect) in a Brisbane host community using a next generation matrix (NGM). Our estimates across a complete-transmission cycle combine all the components listed in *Figure 1* and described in *Box 1*; uncertainty is propagated from fitted statistical sub-models (see *Table 1*).

### Host-vector-host transmission

Estimated host importance changed little between host-to-vector and host-vector-host transmission: humans, birds, possums, horses, and macropods remained in the top cluster of hosts (*Figure 2C*). Despite wide 95% CI of humans that overlapped with birds, possums, horses, and macropods, much of the density distribution of host-vector-host transmission estimates (obtained by propagating uncertainty from all statistical sub-models) for humans falls above that of other species (*Appendix 2—figure 1*). For example, 32% of the distribution of total host-to-host infections for humans is at higher estimates than the upper bound of the 95% CI for birds, the next highest species by median estimate. We estimated that the mosquitoes that would acquire RRV from humans mostly go on to infect humans ('self-infections'), followed by birds, dogs, and to a lesser extent possums. Even when weighting second generation infections by the proportion of hosts that mount a viremic response (i.e., ignoring all sink infections in dogs and thus counting second generation *infectious* hosts only), humans still produce the most second-generation infectious hosts by median estimate, though CI once again overlap with birds, macropods, horses, and possums (*Appendix 2—figure 2*). We predicted that an infected bird (the species with the second highest estimated median) would primarily infect other birds, followed by dogs and humans, respectively (*Figure 2C*).

Because humans are the only species without data from experimental infection studies (titer was measured when infected humans began showing symptoms), we checked the robustness of our results by re-running analyses assuming a host titer duration for humans reflecting only the observed

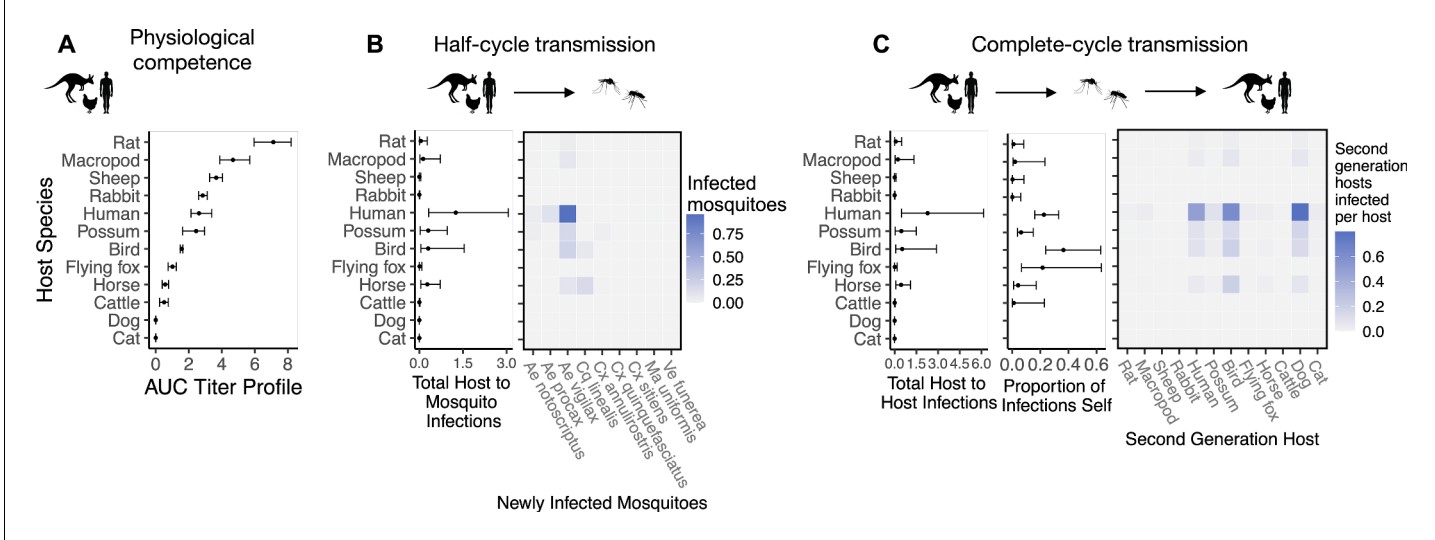

**Figure 2.** The most competent host species for Ross River virus (RRV) transmission in Brisbane change when considering physiological traits alone (**A**) or also considering ecological traits (**B, C**). (**A**) Estimated physiological response of hosts to experimental infection with RRV, summarized using the area under their estimated titer profiles over time (AUC). In all panels, points show median estimates; error bars are 95% confidence intervals (CIs) that combine the uncertainty from all statistical sub-models used to obtain the estimates presented in that panel (see **Figure 1** and **Box 1** for these components). Titer profile AUC is used only to quantify host physiological competence, while time-dependent titer profiles (pictured in **Appendix 1—figure 1**) are used in half-cycle and complete-cycle transmission. The ordering of hosts based on highest (top) to lowest (bottom) physiological competence in A is conserved in B and C to aid visualization of host order changes among panels. (**B**) Host-to-vector transmission; matrices show the median estimated number of vectors infected by each host species, while the points show infection totals (sums across matrix rows), with error bars. (**C**) Host-vector-host transmission. As in B, the matrices show estimated median numbers of next-generation host infections for all host species pairs, while the points show sums across rows of the matrices (left plot) and the proportion of infections in the second generation that are in the same species as the original infected individual (center plot).

human viremic period. Even when human titer duration was reduced, humans remained in the top cluster of hosts (with birds, possums, horses, and macropods) for RRV transmission potential despite an overall lower total number of second-generation infections (**Appendix 2—figure 3**, **Appendix 2—figure 4**). This highlights the robust result that humans likely contribute to the RRV transmission cycle in Brisbane due to their physiological competence, abundance, and attractiveness to competent mosquitoes like *Ae. vigilax* and *Ae. procax*.

## Vector-host-vector transmission

Across a complete vector-host-vector transmission cycle, confidence intervals remained wide for the estimated number of mosquitoes an infected mosquito of each species would infect over its lifetime (**Figure 3C** left panel). Nonetheless, the results suggest that *Cq. linealis*, *Ae. procax*, *Ve. funerea*, *Ae. vigilax*, and *Ma. uniformis* have a much higher maximum transmission potential than *Cx. annulirostris*, *Cx. quinquefasciatus*, *Cx. sitiens*, and *Ae. notoscriptus*.

Importantly, the results pictured in **Figure 3C** calculate second generation mosquito infections conditional on starting with a mosquito exposed to 6.4 $\log_{10}$ infectious units of RRV per mL (the median dose used in experimental infection studies); if it is a rare event that a given mosquito species becomes exposed in the first place, basing mosquito importance on this metric could be misleading. For example, regardless of the species of the originally infected mosquito (rows of the **Figure 3C** matrix), we predict that most second generation infections will be in *Ae. vigilax*, followed by *Ae. procax* and *Cq. linealis* (columns of the **Figure 3C** matrix), because of their abundance and feeding preferences. Similarly, while an individual *Ve. funerea* or *Ma. uniformis* mosquito could potentially have the highest ability for producing second-generation infections in mosquitoes (**Figure 3C**), their rarity (0.27% and 0.14% of the Brisbane mosquito community, respectively; **Supplementary file 2**) means that few second generation infections from any source mosquito occur in *Ve. funerea* or *Ma. uniformis*. Thus, unlike *Ae. vigilax*, *Ae. procax*, and *Cq. linealis*, the rare

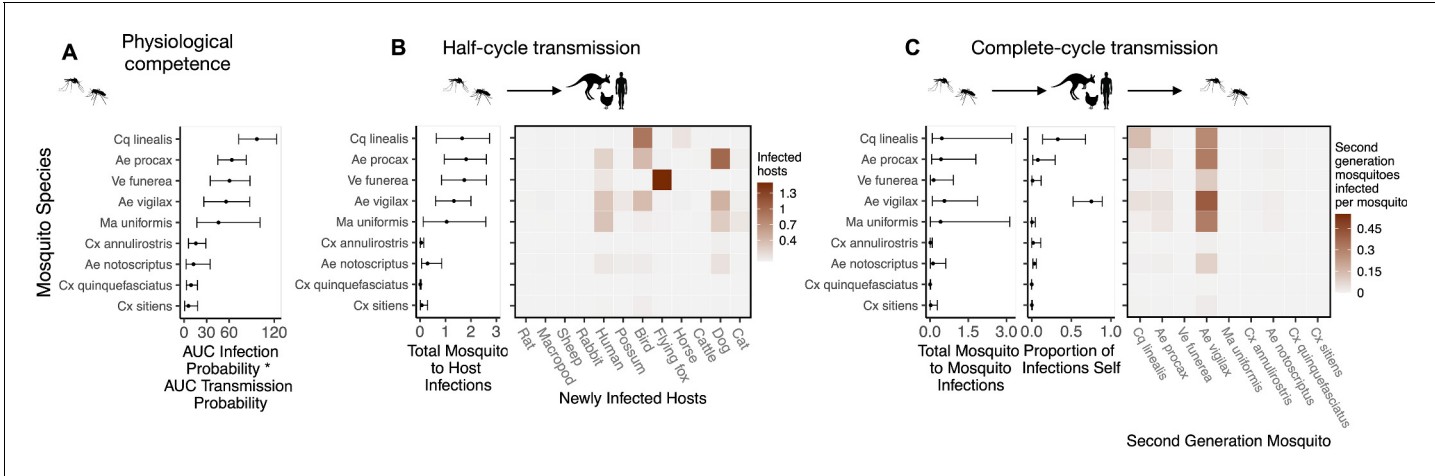

**Figure 3.** Ross River virus (RRV) transmission capability of Brisbane mosquitoes remained consistent when considering physiological traits alone (**A**) or also considering ecological traits (**B, C**). (**A**) Physiological response of mosquitoes to experimental infection with RRV, summarized using the area under (AUC) of their estimated infection probability versus dose curves multiplied by the area under their transmission probability versus time curves. Points show median estimates; the error bars in each panel are 95% confidence intervals (CIs) that combine the uncertainty from all statistical sub-models used to obtain the estimates presented in that panel (see *Figure 1* and *Box 1* for these components). AUC is used only to quantify mosquito physiological competence; raw infection and transmission profiles (pictured in *Appendix 1—figure 3* and *Appendix 1—figure 4*, respectively) are used in calculations of half-cycle and complete-cycle transmission. The ordering of vector species based on highest (top) to lowest (bottom) physiological competence in A is conserved in B and C to aid visualization of vector order changes among panels. (**B**) Vector-to-host transmission; matrices show the median numbers of hosts infected by each vector species, while the points show infection totals (sums across matrix rows), with error bars. (**C**) Vector-host-vector transmission. As in B, the matrices show median numbers of next-generation vector infections for all vector species pairs, while the points show sums across rows of the matrices (left plot) and the proportion of infections in the second generation that are in the same species as the original infected individual (center plot).

mosquitoes *Ve. funerea* or *Ma. uniformis* are very unlikely to play an important role in RRV transmission over multiple generations in this ecological context.

## Multiple generations of transmission

To estimate which host and mosquito species drive RRV spread as it invades a naive host population, we approximated transmission over five complete RRV life cycles using the next-generation matrix (NGM) approach to calculate transmission in discrete time steps where each time step represents a complete cycle of transmission. Simulating the spread of infection over multiple generations, starting with one initially infected human in an otherwise susceptible vertebrate population in Brisbane, shows that infections tend to propagate through humans, birds, dogs, and horses (median estimates: *Figure 4*; estimates with uncertainty: *Appendix 2—figure 5*). Overall, while infection does circulate largely in the broader vertebrate community (as opposed to continuously cycling between a small subset of vectors and hosts), we estimated that at the beginning of an epidemic in Brisbane, many infections would occur in humans and birds, a moderate number in horses, and many sink infections in dogs. These new infected individuals (apart from dogs and cats) continue to spread infection in the community, and already by the third generation of infection, the most dominant pathways of transmission have converged to birds infecting other birds, humans infecting other humans, humans infecting birds, horses infecting humans, and 'wasted' transmissions from both humans and birds to dogs, a dead-end host (*Figure 4* Generation 3).

Starting with an initial infection in a *Ma. uniformis* mosquito (to illustrate the effect of beginning with an infection in a rare species), the multi-generation approximation shows that after only a single generation the framework predicts that the majority of infected mosquitoes will be *Ae. vigilax* and *Ae. procax*, and to a lesser extent *Cq. linealis* and *Cx. annulirostris* (median estimates: *Figure 4*; estimates with uncertainty: *Appendix 2—figure 6*), which mirrors the results in *Figure 3C*. Despite the potentially high competence of *Ma. uniformis*, their rarity in the Brisbane mosquito community causes them to participate little in sustained community transmission. After only three generations, we predicted that most transmission of RRV in Brisbane was occurring from *Ae. vigilax*, *Ae. procax*,

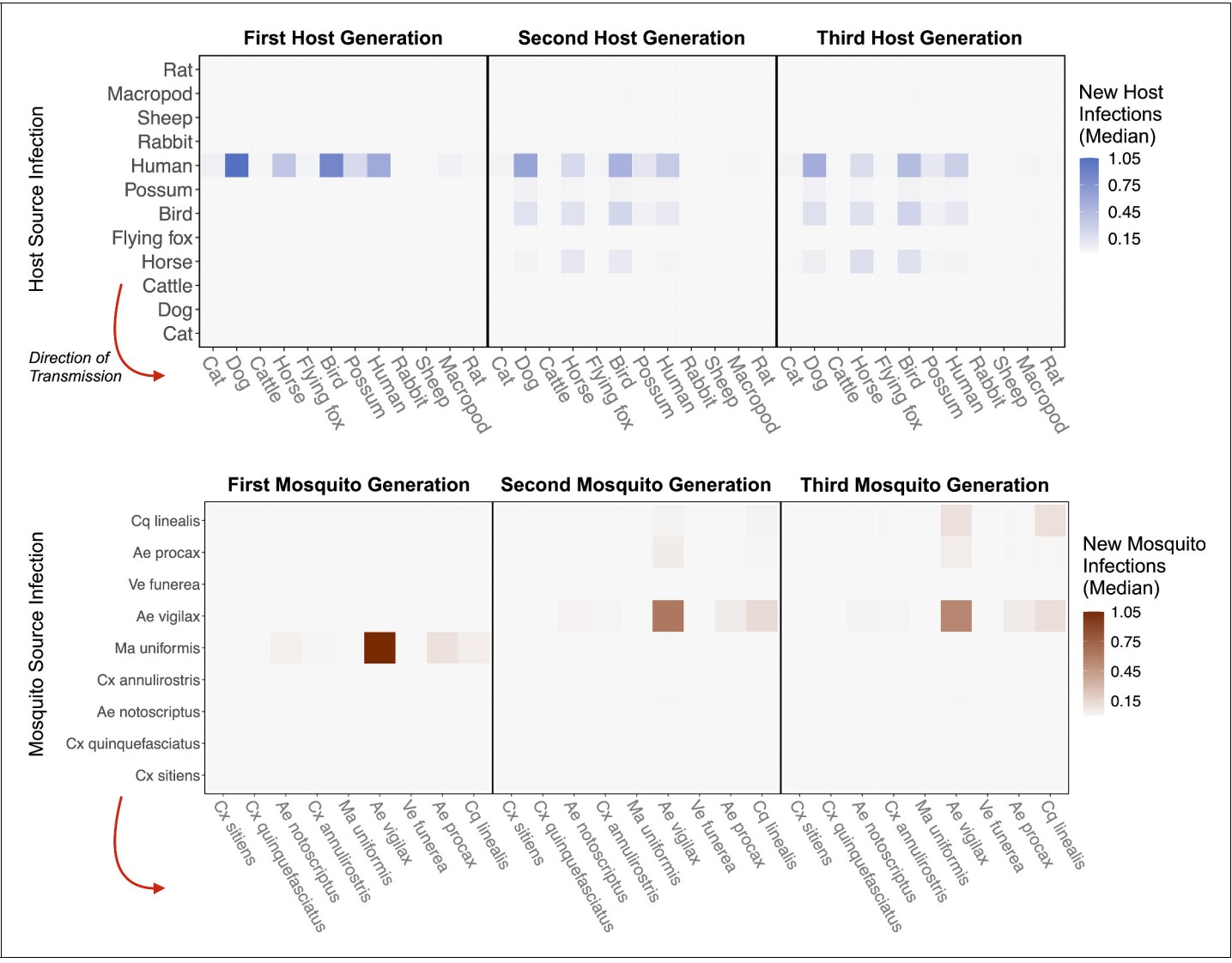

**Figure 4.** RRV epidemic dynamics propagate through initially naïve host and vector communities. Epidemics are simulated in two ways: transmission in the host community resulting from an initial infection in a human (top row), or transmission in the mosquito community arising from a source infection in a *Ma. uniformis* mosquito (bottom row). Each matrix cell contains the median estimated number of new infections in a given species (columns) arising from all infected individuals of a given species in the previous generation (rows). The red arrow shows the direction of infection. We show generations 1–3 here to illustrate how quickly infections propagate through the community and converge on dominant transmission pathways, by generation 3. Uncertainty in the number of new infections in each host and mosquito species over five generations is shown in *Appendix 2—figure 5* and *Appendix 2—figure 6*, respectively.

and *Cq. linealis*; the dominance of these three species can be seen in *Figure 4* by the large number of pairwise transmission events between them.

## Discussion

Motivated by a practical need to identify the relative importance of hosts and vectors for zoonotic arboviral transmission, we developed a nested approach that incorporates existing data, uncertainty, and the complex, dynamic interactions that underpin the transmission of multi-host, multi-vector pathogens. We applied this approach to RRV transmission in Brisbane, which is thought to have multiple transmission cycles (*Stephenson et al., 2018*; *Claflin and Webb, 2015*), and contributes a significant public health burden (*Jansen et al., 2019*). Our approach highlights how species importance changes across physiological and ecological drivers of transmission across half, complete, and

multiple generations of transmission cycles, thus isolating the factors that contribute most to vector or host importance.

## Physiology meets ecology: changes in species importance

The first aim of this study was to characterise which hosts and vectors had high physiological competence for RRV. Species must be able to acquire and propagate the virus to be an important host or vector. Our results corroborate some of what has been previously reported (*Stephenson et al., 2018*; *Harley et al., 2001*), but also generated some surprising results. The strong physiological competence of macropods has long been acknowledged, while cats and dogs have never been considered to play a role as hosts; our research supported both of these ideas. By contrast, horses, which occasionally develop high viremia in response to RRV infection and have been previously considered a moderately competent host (described in *Stephenson et al., 2018*), have low physiological competence on average because less than 15% of exposed horses develop a viremic response when infected. Conversely, humans, which have not been considered important for local transmission, had a moderate-to-high physiological competence following infection with RRV (*Figure 2A*). For vectors, RRV has long been considered a generalist virus, capable of persisting across climates and habitats within Australia; our result that no single species was dominant in its physiological competence supports this view.

Physiological competence alone, without ecological data, provides an incomplete picture of transmission and can be misleading. For example, a host's physiological competence is of little importance if that host is rare or adopts behaviors that prevents exposure (*Downs et al., 2019*). Further, mosquito feeding preferences can drive pathogen transmission more strongly than host competence (*Simpson et al., 2012*). There are many documented circumstances in which species that are highly competent for transmission under controlled conditions play a minor role in community transmission (*Levin et al., 2002*; *Marm Kilpatrick et al., 2006*), or conversely, where species with apparently low competence in laboratory studies are highly important for transmission in nature (*Brady et al., 2014*; *Brook and Dobson, 2015*). We found the former to be the case for RRV hosts across half and complete transmission cycles. For example, we estimated that humans contributed more mosquito infections (*Figure 2B*) and second generation host infections (*Figure 2C*) than the most physiologically competent species (rats, sheep, and macropods; although human 95% CI overlapped that of macropods). There are longstanding debates within disease ecology surrounding how ecological interactions moderate disease dynamics, for example, through dilution effects (*Johnson and Thieltges, 2010*) and zooprophylaxis (*Donnelly et al., 2015*). The nested approach is useful for identifying specific mechanisms because it analyzes transmission as a step-wise process with increasing ecological complexity by integrating different forms of trait data. Specifically, the results from a half transmission cycle represent the pairwise interactions between host and vector species. For example, a physiologically competent host with low community competence based on host-to-vector cycles (for RRV this includes rats, sheep, and rabbits) occurs due to low rates of contact between this host and vectors with a high infection probability. By contrast, a host with low competence across a complete transmission cycle, but high host-to-vector transmission competence, would reflect more on the transmission ability of the vectors that host infects. By separating transmission in this way, we can examine the contribution each trait makes to species importance and test hypotheses such as whether it is more important for a host to infect a greater number and diversity of vectors, or fewer, more competent vectors.

In our study, different ecological drivers likely underpin the importance of humans and birds, the two species with the highest median estimates for complete-cycle transmission (*Figure 2*, *Appendix 2—figure 1*). For example, when compared to all other hosts, humans had the highest susceptible population (contributing 66% of the total community abundance, with less than 14% seropositivity). This, in combination with their moderately-high physiological competence (*Figure 2B*) contributes to their overall importance. These factors are more important than other ecological drivers. For example, although humans infect a large number of moderately competent vectors (*Ae. vigilax* and *Ae. procax*; *Supplementary file 3*), the mosquito feeding patterns potentially limit human importance because many of the mosquitoes reported to feed on humans have lower competence for RRV (such as *Cx. annulirostris* and *Ae. notoscriptus*). That being said, the number of *Ae. vigilax* that humans infect (*Figure 2B*) suggests that a potentially fruitful path for reducing human infections is vector control of *Ae. vigilax* populations, which is already one of the primary

targets of mosquito control operations in Brisbane (*Brisbane City Council,, 2019*). In contrast, birds were estimated to be only approximately 5% of the host community composition and almost a third were seropositive, further reducing the total number of susceptible individuals. Despite this relative scarcity, birds were highly important in the half and complete transmission cycles. This high importance is likely driven by the strong feeding association with the highly physiologically competent mosquito *Cq. linealis* rather than birds' physiological competence or abundance.

## Transmission pathways of RRV in Brisbane

Moving beyond single transmission cycles, when we approximate transmission through the Brisbane community over five generations (approximately the transmission season: *Australian Govt. Dept. of Health, 2020*), we estimate that infection spreads widely through the community, with the largest number in humans, birds, dogs, and horses. The physiologically competent, abundant, and generalist feeder *Ae. vigilax* plays an important role in this propagation. Despite large uncertainty, our findings for RRV transmission cycles in Brisbane point to two overlapping transmission cycles: an enzootic cycle, characterized primarily by transmission between birds and *Cq. linealis*, and a domestic cycle characterized by human-to-human infections facilitated by *Ae. vigilax* and *Ae. procax*. These two cycles are linked by these feeding generalists, which transfer infection between birds and humans. Within each of these overlapping cycles, dogs play a diluting role by absorbing infectious bites as they are not able to transmit RRV.

Multiple transmission cycles for RRV have long been hypothesized (*Harley et al., 2001*), yet no previous studies have implicated the species involved in these cycles or quantified their contribution to transmission. Humans and birds have been greatly understudied as potential hosts of RRV, yet unlike marsupials, they persist across the geographic distribution of RRV. Despite frequent detection of RRV in major metropolitan centers (*Claflin and Webb, 2015*), the potential for humans to contribute to endemic transmission (as opposed to epidemic transmission: *Rosen et al., 1981*; *Aaskov et al., 1981*) has empirically been understudied. Although our predictions provide some support for the importance of these understudied pathways, because we were unable to model seasonal changes in vector abundance or the correlated seasonal changes in human RRV cases in Brisbane (which generally peak in late summer through early autumn: *Australian Govt. Dept. of Health, 2020*), more modeling and empirical work is needed. Hopefully our identification of multiple transmission pathways will allow for future research to formulate hypotheses for RRV seasonality. For such work data would need to be collected across seasons to distinguish the role of seasonality and the timing/drivers of spillover that shift transmission from an enzootic to domestic cycle.

The vectors identified in Brisbane transmission cycles, *Ae. vigilax*, *Ae. procax* and *Cq. linealis*, are recognised as important vectors for RRV and are regularly targeted in vector control programs. However, we predicted that *Cx. annulirostris* and *Ae. notoscriptus* are less competent vectors, although they are often cited as key RRV vectors in Brisbane (*Kay and Aaskov, 1989*; *Russell, 1995*; *Watson and Kay, 1998*). The evidence in favor of *Cx. annulirostris* as a vector is that RRV is frequently detected in wild-caught individuals, and that abundance has been high during previous outbreaks of RRV (*Jansen et al., 2019*). RRV has also been isolated from *Ae. notoscriptus* during outbreaks in Brisbane (*Ritchie et al., 1997*); however, the species had relatively low abundance in this study, and low transmission ability (*Appendix 1—figure 4*) in comparison to other potential vectors. This suggests a new hypothesis that *Cx. annulirostris* and *Ae. notoscriptus* are secondary RRV vectors (capable of playing a supplemental role in transmission but unable to maintain an epidemic) to other species such as *Ae. vigilax* which are primary RRV vectors (capable of starting and maintaining epidemics). Although novel for RRV, the distinction between primary and secondary vectors has been made for other arboviruses (*Turell et al., 2005*). Finally, the isolation of RRV from wild caught mosquitoes demonstrates that a particular species is infected with the virus, it is incomplete evidence for mosquito species' specific role in virus transmission. Even if found infected in the field, the lower transmission capability of *Cx. annulirostris* or *Ae. notoscriptus* relative to *Ae. vigilax*, *Ae. procax* and *Cq. linealis* means that the former are likely to transmit infection to fewer hosts than the latter.

## Caveats and uncertainty

It is important to acknowledge a number of caveats with the data and modeling assumptions we used. For physiological competence, experimental studies vary substantially in their methods. We overcame some of this variation by transforming published data into the same viral units between studies (e.g., infectious units were converted to per milliliter: IU/mL). However, not all variation in experimental approaches could be included in our regression model because of data sparsity. Thus, it is possible that some of the variation we attribute to species may in fact be explained by methodology used in different studies. For the ecological data, the methods used to collect species abundance data can also result in bias, as different traps and survey types detect different species (*Brown et al., 2008*; *Lühken et al., 2014*). For example, the species trapped using $CO_2$-baited light traps in this study may not be a true representation of the entire mosquito community in Brisbane. Similarly, vertebrate survey methods are biased against detecting species with cryptic behavior, and thus represent a biased sample of the host community available to host-seeking mosquitoes. While the uncertainty captured in the reported data were propagated through our estimates of competence, unmeasured uncertainty arising due to experimental methods could additionally affect the results. However, compared with approaches that focus solely on a single physiological or ecological data source to infer competence, the approach presented here allows for a more detailed investigation of vector and host competence and their drivers.

There are many potential hosts that are not included in this analysis due to data limitations. As a minimum requirement, host species were only included if they were included in mosquito blood meal field observations, were experimentally exposed to the virus, and were measured for background seroprevalence and abundance in Brisbane. In some instances, to meet these minimum data requirements, species were aggregated by taxonomic group. For example, we averaged the responses of chickens, little corellas, and Pacific black ducks to 'birds' (while a strong simplifying assumption, the clustering of these species' physiological response does provide support for this choice: *Appendix 1—figure 2*). In other instances (such as the potential for koalas to be hosts of RRV), species were unable to be modeled because of an absence of viremia data. Further, we ignore seasonal matching of transmission with host reproduction, ignore duration of host life stages, and either make a snapshot measure of host transmission capability (*Figure 2*, *Figure 3*) or make a simple five-generation approximation that averages across host and vector infectious periods (*Figure 4*). Finally, some hosts and vectors may only be locally important for RRV transmission, as opposed to being important over the entire geographic distribution of the virus. For example, though sheep have high physiological importance, they were not locally important in Brisbane. However, sheep could play a greater role in the maintenance and spillover of RRV in rural areas where they are more abundant and/or where other species of mosquitoes with higher biting affinity for sheep may occur.

For mosquitoes, data sets with the most substantial gaps included host feeding data, physiological transmission capability, and mosquito survival. Blood meal data is difficult to collect, but is very important because feeding patterns enter into the equation twice for vector-host-vector transmission. Limited blood meal counts (*Supplementary file 3*) led to high uncertainty in feeding patterns for many species (e.g. *Ma. uniformis*), which can have a large influence over the width of the 95% CI (*Figure 3C*). Addressing these data gaps is critical for refining vector predictions for RRV, though these data are logistically difficult and costly to obtain. More laboratory experiments on mosquito transmission probability over time, especially for those understudied species that we predict have the potential to be important transmitters would also help to better resolve transmission patterns in the Brisbane community. For example, the 95% CIs for *Ma. uniformis* and *Ve. funerea* are particularly wide, which could place them as either highly important vectors or inefficient vectors. Finally, because we assumed identical survival for all species, with no uncertainty (i.e., survival did not contribute to the widths of the confidence intervals across species), the uncertainty we present is an underestimate. Species-specific field-based mortality rates are a crucial data source that needs to be obtained for more accurate measures of mosquito transmission capability. It is important to note, however, that even in spite of large uncertainty for vector-host-vector transmission (*Figure 3C*), the rarity of many of these mosquito species make them mostly irrelevant when approximating transmission over multiple generations (*Figure 4*, *Appendix 2—figure 6*).

While all of these modeling choices and data shortcomings can influence model outcomes, a clear advantage of the framework is that uncertainty from each statistical sub-model fit to independent

data sets is accounted for in the overall estimates. In doing so, parameters with high uncertainty, such as mosquito feeding preferences or transmission probabilities, can be targeted in future studies to help refine the framework's predictions.

## Applications for other vector borne diseases

This framework can be applied to other vector-borne pathogens in a number of ways. A principal application would be to identify important vectors and hosts for other multi-host, multi-vector pathogens, including RVFV (*Turell et al., 2008*; *Davies and Karstad, 1981*; *Gora et al., 2000*; *Busquets et al., 2010*); WNV (*Kain and Bolker, 2019*), or yellow fever virus (*Rosen, 1958*; *Jupp and Kemp, 2002*), for which competence data exist for several species. For these viruses, our framework and code can be used by substituting data and modifying the underlying statistical sub-models (e.g., titer profiles) to match the dynamics of the pathogen of interest; the subsequent calculations for host and vector competence, half-cycle transmission, and complete-cycle transmission are usable without modification. The generality of this framework and its nested approach can also support (with minimal modification) additional transmission pathways such as vertical transmission (where mosquitoes emerge from immature stages already infected with a given pathogen), or direct verte-brate-to-vertebrate transmission as can occur for some vector-borne diseases such as RVFV (*Wichgers Schreur et al., 2016*) or Zika virus (*D'Ortenzio et al., 2016*).

Secondary applications for this framework could include identifying the largest gaps and uncer-tainties within datasets. This is advantageous because in light of finite resources, model-guided research (*Restif et al., 2012*) can identify the most important data needed to improve predictions for disease emergence and transmission. Another application would be to apply the framework for a single pathogen across space and time, such as across the geographic range of RRV or between sea-sons. This is useful to compare shifts in transmission dynamics, identify hotspots or potential for spill-over. Though our framework has not been developed to predict the timing and peak of epidemic events, it can be used to disentangle the underlying transmission dynamics of vector-borne patho-gens in specific locations, which allows for the development of predictive modeling.

Finally, the generality and multi-phase nature of this framework provide a common language to compare and contrast the transmission dynamics not just within a single pathogen, but also between them. Until now, the highly diverse methods, definitions and data required to characterise vectors and hosts has hindered the ability to make comparisons between pathogens. The integration of mul-tidisciplinary data in this framework is done in a way that could be used to compare host or vector physiological competence and ecological traits for other multi-host, multi-vector pathogens.

## Conclusion

Identifying important vectors and hosts of zoonotic pathogens is critical for mitigating emerging infectious diseases and understanding transmission in a changing world. However, attempts to do so have been hampered by the multidisciplinary datasets required and differing definitions that can alter the importance of a species. Here we developed a nested approach that can be applied to any multi-host, multi-vector pathogen for which some competence data exists. Applying this approach to RRV transmission in Brisbane, we were able to: (a) identify two hosts of potentially high impor-tance that deserve further investigation (humans and birds), (b) two potential transmission cycles (an enzootic cycle and a domestic cycle), and (c) datasets that should be targeted (bloodmeal studies, vector transmission experiments, field-based mosquito survival estimates) to reduce overall uncer-tainty and ultimately increase the future power of the framework. Future studies that aim to identify and quantify the importance of different species in virus transmission cycles must integrate both physiological competence data and ecological assessments to more fully understand the capacity of species to transmit pathogens. The nested approach here provides a tool to integrate these differ-ent datasets while acknowledging uncertainty within each, which could be applied to any multi-host, multi-vector pathogen for which some competence data exists.

## Materials and methods

The methods are presented in three sections to reflect our three focal questions. First, we describe the calculation of host and vector physiological competence. Second, we describe half-cycle (host-to-vector and vector-to-host transmission) and complete-cycle (host-vector-host or vector-host-

vector) transmission. Third, we describe how we use complete-cycle transmission to approximate transmission over multiple generations. We introduce data and calculations for components that are used in multiple transmission metrics (e.g., host virus titer profiles) with the first metric in which they are used.

## Host and vector physiological competence

### Vertebrate hosts: virus titer profiles

We fit host virus titer profiles as continuous functions over time to published data on host vertebrate responses to infection. For each of 15 experimentally infected non-human vertebrate species we extracted the proportion of exposed individuals that developed detectable viremia, their duration of detectable viremia in days, their peak viremia titer, and the unit of measure of this titer (such as median lethal dose ($LD_{50}$), suckling mouse intracerebral injection ($SMIC_{50}$)) (from *Whitehead, 1969*; *Spradbrow, 1973*; *Rosen et al., 1981*; *Kay et al., 1986*; *Ryan et al., 1997*; *Boyd et al., 2001*; *Boyd and Kay, 2002*). All reported viral concentrations were converted to infectious units per millilitre (IU/mL) values, rather than 0.1 mL or 0.002 mL as reported in some studies. Titer data are summarized in *Supplementary file 1* and a summary of these studies' methodological details can be found in *Stephenson et al., 2018*; all data extracted from these publications are available in *Source data 1*.

For non-human species, only means and standard deviations for peak titer and duration of detectable titer were reported. We transformed these summary measures into continuous titer profiles (continuous functions of titer over time that are needed to quantify mosquito infection probability) by modeling titer profiles as quadratic functions of time since infection, based on observed patterns in the data. For human titer profiles, for which experimental infection studies were not available, we used data from one observational study (*Rosen et al., 1981*) that measured titer in humans exhibiting disease symptoms during an outbreak in the Cook Islands in 1980. Details on how we constructed continuous titer curves, with uncertainty, for all hosts are available in Appendix 1; for raw human titer data see *Source data 2*. In *Appendix 1—figure 1* we show 95% confidence intervals (CI) for each of the hosts' quadratic profiles generated from this procedure with the summary values of peak and duration of titer extracted from the literature overlayed. To quantify host physiological competence we summarized the titer profiles into a single metric using the area under the curve (AUC) of the time-dependent titer curves. We use AUC because it simultaneously captures both titer magnitude and the duration of detectable titer (the host's infectious duration). AUC is used only to summarize host competence; raw time-dependent titer values are used to calculate mosquito infection. The AUC for the fitted titer profiles (*Appendix 1—figure 1*) are shown in *Appendix 1—figure 2*.

### Mosquito vectors: infection and transmission probability

We fit mosquito infection probabilities and mosquito transmission probabilities using published data from laboratory experimental exposure of mosquitoes to RRV. From experimental infections of mosquitoes we collected information on the infectious dose they were exposed to, the number of mosquitoes receiving an infectious dose, the proportion of mosquitoes that became infected, the proportion of mosquitoes that went on to become infectious (i.e., transmitted the virus), and the time it took for mosquitoes to become infectious (the extrinsic incubation period) (from *Kay et al., 1979*; *Kay et al., 1982*; *Kay, 1982*; *Kay et al., 1982*; *Ballard and Marshall, 1986*; *Fanning et al., 1992*; *Vale et al., 1992*; *Wells et al., 1994*; *Doggett and Russell, 1997*; *Watson and Kay, 1998*; *Jennings and Kay, 1999*; *Ryan et al., 2000*; *Doggett et al., 2001*; *Jeffery et al., 2002*; *Kay and Jennings, 2002*; *Jeffery et al., 2006*; *Webb et al., 2008*; *Ramírez et al., 2018*). Mosquito infection and transmission data are summarized in *Supplementary file 2*; raw data files are included as *Source data 3* and *Source data 4*, respectively.

We modeled both mosquito infection probability (the proportion of all experimentally exposed mosquitoes with virus detected in their bodies) and transmission probability (the proportion of all experimentally exposed mosquitoes with virus detected in their saliva, measured via feeding on a susceptible vertebrate species or using an *in vitro* method of saliva collection) using generalized linear mixed effects models (GLMM) with Binomial error distributions, fit in R using the package lme4 (*Bates et al., 2015*). For each model, the proportion of mosquitoes infected or transmitting was

taken as the response variable and the total number exposed to infection was used as weights; species were modeled using random effects. For additional details see the Supplemental Methods. Fitted infection probability curves for all mosquito species for which we gathered data—those found in Brisbane and elsewhere in Australia—are shown in *Appendix 1—figure 3*; transmission probability curves are shown in *Appendix 1—figure 4*. To quantify mosquito physiological competence we summarized mosquito infection and transmission probabilities into a single metric using the area under the curve (AUC) of the dose-dependent infection curve multiplied by the area under the curve (AUC) of the time-dependent transmission curve. AUC is used only to summarize mosquito competence; raw probability values are used to calculate the probability a mosquito becomes infected when feeding on an infected host (given the titer in that host) and the probability they are able to transmit to a susceptible host (given the number of days post infection that the feeding occurs). The AUC for the fitted infection probability (*Appendix 1—figure 3*) and transmission probability (*Appendix 1—figure 4*) curves are shown in *Appendix 1—figure 5* and *Appendix 1—figure 6*, respectively.

## Half-cycle and complete-cycle transmission

Both half-cycle (host-to-vector and vector-to-host) and complete-cycle (host-vector-host and vector-host-vector) transmission nest host and vector physiological competence in an ecological context (*Figure 1*). To quantify each of these metrics we used a next-generation matrix (NGM) model (*Diekmann et al., 1990*; *Hartemink et al., 2009*), which, for a vector-borne disease, requires the construction of two matrices of transmission terms. The first matrix (denoted HV, where bold terms refer to matrices) contains species-specific host-to-vector transmission terms, which we write with hosts as rows and vectors as columns. The second matrix (VH) contains vector-to-host transmission terms and has vectors as rows and hosts as columns. Cells of HV and VH contain the expected average number of infections between pairs of species over the whole infectious period of the infector (host in HV, vector in VH); each pairwise transmission term is a function of host and vector physiological competence as well as ecological factors. Row sums of HV give the total number of vectors (of all species) infected by each host (total host-to-vector transmission); similarly row sums of VH give the total number of hosts (of all species) infected by infectious vectors.

We calculate the total number of individuals of each mosquito species $j$ that a host species $i$ infects over its infectious period $d$ (which gives entry [i, j] of HV) as:

$$Iv_{ij} = \sum_{d_i=1}^{9} (p_j | \theta_{id_i}) \cdot \omega_i \cdot \phi_{ij} \cdot \sigma_j \cdot \frac{\beta_{ij}\alpha_i}{\sum_{i=1}^{I} \beta_{ij}\alpha_i}, \qquad (1)$$

where $p_j | \theta_{id_i}$ is the probability that a susceptible species of mosquito ($j$) would become infected when biting host $i$ on day $d_i$ when it has titer $\theta_{id_i}$. We model infection over a period of 9 days for all host species given that the estimated titer of all host species is predicted to be undetectable by 9 days, equating to a very small mosquito infection probability (*Appendix 1—figure 1*). The proportion of individuals of species $i$ that manifest an infection with detectable titer $\theta_{id_i}$ is given by $\omega_i$, while $\phi_{ij}$ is the number of susceptible mosquitoes of species $i$ per host species $j$, $\sigma_j$ is the daily biting rate of mosquito species $j$, and $\frac{\beta_{ij}\alpha_i}{\sum_{i=1}^{I} \beta_{ij}\alpha_i}$ is the proportion of all mosquito species $j$'s bites on host species $i$, which is jointly determined by the relative abundance of host $i$ ($\alpha_i$) and the intrinsic feeding preference of mosquito $j$ on host $i$ ($\beta_{ij}$) (details given in *Mosquito feeding behavior* below). *Equation 1* assumes no species specific host-by-mosquito interactions for infection probability; mosquito infection probability is uniquely determined by the level and duration of titer within a host (i.e., a dose-response function of host titer). The only direct evidence against this assumption that we are aware of is an example where more *Cx. annulirostris* became infected when feeding on a bird than on a horse despite there being a lower viremia in the bird (*Kay et al., 1986*).

The total number of individuals of each host species $i$ that a mosquito of species $j$ infects over its infectious period $r_j$ (which gives entry [j, i] of VH) is given by:

$$Ih_{ji} = \sum_{r_j=1}^{38} p_{ir_j} \cdot \eta_j \cdot \lambda_{jr_j} \cdot \sigma_j \cdot \frac{\beta_{ij}\alpha_i}{\sum_{i=1}^{I} \beta_{ij}\alpha_i}, \qquad (2)$$

where $p_{ir_j}$ is the probability an infected mosquito of species $j$ transfers infection to a given

susceptible host by bite on day $r_j$ of their infectious period, $\lambda_{jr_j}$ is the probability of survival of mosquito species $j$ until day $r_j$, $\sigma_j$ is the daily biting rate of mosquito species $j$, and $\frac{\beta_{ij}\alpha_i}{\sum_{i=1}^{I}\beta_{ij}\alpha_i}$ is the proportion of all mosquito species $j$'s bites on host species $i$. We calculate mosquito-to-host transmission over 38 days given that we assume mosquitoes do not survive longer than 38 days (see *Mosquito survival* below).

The key differences between the host-to-vector (HV; $Iv_{ij}$) and vector-to-host (VH; $Ih_{ji}$) transmission matrix entries are two-fold. First, HV assumes that host infectivity is titer- and time-dependent and depends on mosquito density per host; conversely, VH assumes that mosquito infectiousness is titer-independent (dose-independent) but time-dependent and depends on daily mosquito survival and host species relative abundance. Second, for HV we assume a single infected host of a given species enters into a community of susceptible mosquitoes, while for VH we assume that a single mosquito of a given species becomes exposed to a dose of 6.4 $\log_{10}$ infectious units per mL (the median dose used across all mosquito infection studies) and then enters a host community with empirically estimated background host immunity (from *Doherty et al., 1966*; *Marshall et al., 1980*; *Vale et al., 1991*; *Boyd and Kay, 2002*; *Faddy et al., 2015*; *Skinner et al., 2020*; see *Supplementary file 1* and *Source data 7* for sample sizes and the proportion of each host testing seropositive for RRV). The primary similarity between these matrices is that mosquito biting rate, host abundance, and mosquito feeding preference ($\sigma_j$ times the fraction of $\alpha$ and $\beta$ terms) are used in both matrix calculations as the components that control the contact rate between infected hosts and susceptible mosquitoes (VH) or infected mosquitoes and susceptible hosts (VH).

Complete-cycle transmission is calculated using the matrix product of HV and VH, which is commonly referred to as the 'who acquires infection from whom' matrix (*Schenzle, 1984*; *Anderson and May, 1985*; *Dobson, 2004*). Specifically, using HV*VH gives $G_{HH}$, in which each cell describes the total number of pairwise host-vector-host transmission events, assuming a single infected host appears at the start of its infection in an otherwise susceptible host population. Likewise, using VH*HV gives $G_{VV}$, in which each cell describes the total number of pairwise mosquito-to-mosquito transmission events, assuming a single infected mosquito appears at the start of its infectious period in an otherwise susceptible mosquito population. Row sums of $G_{HH}$ give the total number of new host infections in the second generation that originate from single source infections in each host species (total host-vector-host transmission), or the total number of mosquito-to-mosquito transmission events in the case of $G_{VV}$. Column sums of $G_{HH}$ or $G_{VV}$ give the total number of newly infected individuals of each host or mosquito species arising from one infection in each host or mosquito, respectively. These properties can be used to find, for example, dead-end hosts (i.e., 'diluters'; *Schmidt and Ostfeld, 2001*), which would be captured by host species with a small row sum and large column sum in GHH. Further, *Diekmann et al., 1990* show that the dominant eigenvalue of either $G_{HH}$ or $G_{VV}$ describes $\mathcal{R}_0$, the typical number of secondary cases, resulting from pathogen transmission in the heterogeneous community whose pairwise transmission dynamics are described in HV and VH.

We estimated each of the parameters of HV and VH using either statistical sub-models fit to empirical data or directly from empirical data taken from the literature. Uncertainty from all statistical sub-models was propagated into the calculations of HV and VH in one of three ways: (1) titer: by simulating 1000 titer curves given the uncertainty in peak titer and duration of titer in the published data sources (see Supplemental Methods); (2) mosquito infection probability and mosquito transmission probability: by constructing density distributions using the means and variance-covariance matrix of the estimated coefficients assuming univariate or multivariate normality (using 1000 samples; see *Kain and Bolker, 2017*, *Kain and Bolker, 2019* for two examples using this method of uncertainty propagation in similar frameworks); (3) mosquito feeding behavior: using the estimated Bayesian posterior. We do not consider uncertainty for those framework components that rely on raw data (the proportion of hosts that mount a viremic response, host and mosquito relative abundance, and host seroprevalence) or point estimates (mosquito to host ratio, mosquito biting rate, and mosquito survival). Thus, the 95% CIs we present contain uncertainty from fitted statistical models but do not account for the full uncertainty. All of our framework's parameters, the data used to parameterize all sub-models within the framework, and methods of uncertainty propagation are listed in *Table 1*. Details on vertebrate host and mosquito abundance, mosquito survival, and mosquito feeding behavior are described below.

## Vertebrate host abundance

Vertebrate abundance data for Brisbane was calculated from a variety of sources including published literature and technical reports (see *Supplementary file 1* and *Source data 5*). Data on livestock species (cattle, sheep, horses) and humans arose from technical reports undertaken by agricultural and government agencies (*Australian Bureau of Statistics, 2018*; *Meat and Livestock Australia, 2019a*; *Meat and Livestock Australia, 2019b*; *Ward et al., 1996*). Cat and dog abundance was derived from a general pets per human ratio from a technical report (*Animal Medicines Australia, 2019*), and scaled to the human population in Brisbane. Abundance for wildlife was derived either from citizen science reports (birds, possums and macropods: *Australian EPA, 2019*), or published fauna surveys undertaken in Brisbane (flying foxes: *Queensland Government, 2020*; rats, rabbits: *Skinner et al., 2021*). Host abundance was calculated as a measure of density within Brisbane (hosts per km$^2$). We used the relative densities of each of these species as reported in these sources as the species' proportions in our community for our analysis.

## Mosquito abundance

Mosquito relative abundances were estimated for Brisbane by combining data from mosquito surveys (requested from the Brisbane City Council mosquito surveillance program). In brief, Brisbane City Council operates weekly $CO_2$ plus 1-octen-3-ol baited Centers for Disease Control (CDC)-style light traps across 10 sites in Brisbane. Traps are set 1.5 m off the ground before dusk, and collected just after dawn the following morning. Any trapped mosquitoes are stored in $-20°C$ until identification to species level by a single person. This data is not publicly available, but has been analyzed and described in *Skinner et al., 2021*. Mosquito abundance from these surveys was calculated as an average weekly total during peak mosquito season (October to May). Mosquito species abundance data was also supplemented with the results of analyses of the vertebrate host origin of mosquito blood meals presented in previous published studies (*Ryan et al., 1997*; *Kay et al., 2007*; *Jansen et al., 2009*). Mosquito abundance data is summarized in *Supplementary file 2*; raw data is available in *Source data 8*.

We used the observed proportion of each mosquito species detected in these surveys as the proportion of that species in our community for our analysis, which assumes that the observed species proportions are unbiased predictors of their true proportions. Because the number of mosquitoes per host (*Equation 1*: $\phi$) is needed to calculate the absolute number of mosquitoes an infected host would infect, we multiplied the relative abundances of mosquitoes by 40 (our assumed value for overall raw number of mosquitoes per host in the community). While this may be an over- (or under-) estimate of the true value in Brisbane, because this value is only a scalar in the NGM framework it will only affect the magnitude of estimates and not the relative estimates among species.

## Mosquito survival

Survival data (either field or laboratory derived) for the mosquito species present in Brisbane, Australia, is not available for most species. For this reason, we modeled mosquito survival as being identical for all species. Specifically, we used an exponential decay model for mosquito survival using a daily survival probability that is half of the daily maximum survival rate of *Cx. annulirostris* (calculated as 1/lifespan) measured in optimal laboratory conditions (from *Shocket et al., 2018* who used data from *McDonald et al., 1980*, which may over-estimate survival rates in nature). However, we assume that mosquito survival probability falls to zero after day 38.

## Mosquito feeding behavior

We modeled the observed blood meals in wild-caught mosquitoes (the number of blood fed mosquitoes and the source of the blood meals) as arising jointly from the abundance of each host in the community and each mosquitoes' intrinsic feeding preference on each host species (the latent variable that we model here). Data was extracted from published blood meal surveys specific to Brisbane (from *Ryan et al., 1997*; *Kay et al., 2007*; *Jansen et al., 2009*); mosquito blood meal data is summarized in *Supplementary file 2* and *Supplementary file 3*; raw data is available in *Source data 6*. Specifically, we modeled the number of blood meals a mosquito of species $j$ obtains from host species $i$ ($\delta_{ij}$) as:

$$\delta_{ij} \sim Multi(N, \frac{\beta_{ij}\alpha_i}{\sum_{i=1}^{I} \beta_{ij}\alpha_i}),$$ (3)

where $\delta_{ij}$ is a multinomially distributed random variable (the extension of the binomial distribution for greater than two outcomes) with probability equal to the intrinsic preference of mosquito $j$ for host species $i$ ($\beta_{ij}$), weighted by the abundance of host species $i$ ($\alpha_i$), relative to all host species in the community (sum over all host species in the denominator). Written in this way, $\beta_{ij}$ is the ratio of the proportion of bites mosquito species $j$ takes on host species $i$ relative to biting host species $j$ in proportion to their abundance in the community (which would occur if a mosquito were biting randomly). We fit this multinomial model in a Bayesian context in Stan (*Carpenter et al., 2017*), interfaced with R using the package rstan (*Stan Development Team, 2020*). For details on the fitting of this Bayesian model see Appendix 1; the full Stan model is also available in the GitHub repository hosting the code: *Kain, 2021a*.

## Tailoring the model to the Brisbane community

One difficulty with the integration of diverse data types is variation in the biological scale at which these data are collected. For our model, vertebrate host types are recorded at different taxonomic levels across data sets (e.g. laboratory infection experiments are conducted at the species level while mosquito blood meal surveys report identification of the blood meal host source at a taxonomic level ranging from species through to higher level classification such as class or family). In order to integrate the predictions from our individual sub-models fit to single data types (e.g. infection experiments and blood meal surveys) to parameterize HV and VH, and thus draw inference on the importance of different hosts and mosquitoes in RRV transmission in Brisbane, Australia, we made three simplifying assumptions. First, we averaged each mosquito's infection probability when biting 'birds' (the taxonomic level available for blood meal data) for the three species of birds with a measured viremic response (Pacific black duck: *Anas superciliosa*, domestic chicken: *Gallus gallus domesticus*, and little corella: *Cacatua sanguinea*) and 'macropods' for the two macropod species with a measured viremic response (agile wallaby: *Macropus agilis* and eastern grey kangaroo: *Macropus giganteus*). This averaging implicitly assumes (in the absence of species-level information) that all birds and all macropods respond identically to infection. Although a strong simplifying assumption, the three bird species have very similar viremic responses, as do the two macropod species (*Appendix 1—figure 2*). Second, we summed all individuals of all bird species and all macropod species recorded in the Brisbane host surveys in order to calculate the relative abundance of each of these host types to match the aggregation of titer profiles (see *Supplementary file 1* for the relative abundance of each host type in Brisbane). Finally, we retained only nine mosquito species for which we had both abundance data and blood meal data (*Supplementary file 2*), although this excludes many potentially relevant mosquito species, the nine species we retained account for 90% of the Brisbane mosquito community according to our abundance data (*Supplementary file 1*). Our inference on host importance in Brisbane, Australia is thus focused on the following host groupings: birds, cats, cattle, dogs, flying foxes, horses, humans, macropods, possums (namely Brushtail possums *Trichosurus vulpecula*), rats, rabbits, and sheep. We consider the importance of the following mosquito species: *Ae. notoscriptus*, *Ae. procax*, *Ae. vigilax*, *Cq. linealis*, *Cx. annulirostris*, *Cx. australicus*, *Cx. quinquefasciatus*, *Cx. sitiens*, *Ve. funerea*, and *Ma. uniformis*.

## Multi-generation approximation

We approximated how RRV would spread in a naive host and mosquito community at the start of an epidemic to highlight which infection pathways drive transmission as RRV invades. To approximate epidemic transmission, we used the next-generation matrix (NGM) approach to calculate the progression of the disease in discrete time steps where each time step represents a complete cycle of transmission. Because this method relies on the total number of mosquitoes infected over a host's entire infectious period (9 days) and the total number of hosts infected by a mosquito over its entire lifespan (38 days; weighted by their probability of surviving over this period), it approximates how epidemics would propagate if pathogen transmission occurred in discrete generations, rather than continuously in overlapping generations. It is therefore a simplification that does not fully represent time-dependent epidemic dynamics. We use this simulation simply to highlight the host and

mosquito species that would experience the most infections early in an epidemic (given by the total transmission potential across both a host's and mosquito's infectious period).

Specifically, we first calculated the number of hosts of each species that would become infected starting with a single infected host individual of one species using $G_{HH}$. To calculate which hosts would become infected in the next generation, we then used $G_{HH}$ starting with the individuals infected from the previous step. We repeated this process over only five generations to avoid modeling transmission over a longer period than one transmission season in Brisbane. By using the Brisbane community in which RRV is endemic, we use this analysis as an illustrative example of disease emergence and not to provide specific predictions for RRV emergence in any specific new location with no prior exposure to RRV. To estimate how infection spreads in the mosquito community, we used a similar approach, but instead started with one infected mosquito and used $G_{VV}$. As with host-vector-host transmission using $G_{HH}$, while this strategy provides only a coarse approximation of transmission over time by assuming discrete generations of infection, it is useful for revealing important pathways of transmission and identifying species that remain important transmitters over multiple generations without the need to parameterize a dynamic, continuous-time epidemic model.

All data used in this study are uploaded as Source Data files. All codes are hosted on GitHub (https://github.com/morgankain/RRV_HostVectorCompetence, copy archived at swh:1:rev: be7e87c3c4c8af0420a8dd42cdcff5586fdbad90): *Kain, 2021a*; *Kain, 2021b*.

## Acknowledgements

We thank the Mordecai and McCallum labs for feedback on model construction and our presentation of results. We further thank the Mordecai lab for feedback on our first draft of the manuscript. We thank Leon Hugo (QIMR) for advice relating to vertebrate titre profiles, John Mackenzie for advice on human titres, Charles El-Hage for advice on horse populations, and Cameron Webb for advice on mosquito ecology. We also thank the Brisbane City Council for mosquito abundance data. Finally, We thank our reviewers, whose comments improved our manuscript. EAM was supported by the National Science Foundation and the Fogarty International Center (DEB-1518681 and DEB-2011147), the King Center for Global Development, and the Terman Award. EAM, MPK, and EBS were supported by the National Institute of General Medical Sciences (R35GM133439). MPK was supported by the Natural Capital Project. AVDH states that: 'The opinions, interpretations and conclusions are those of the author and do not necessarily represent those of the organization'.

## Additional information

### Funding

| Funder | Grant reference number | Author |
| --- | --- | --- |
| National Science Foundation | DEB-1518681 | Erin A Mordecai |
| National Science Foundation and Fogarty International Center | DEB-2011147 | Erin A Mordecai |
| National Institute of General Medical Sciences | R35GM133439 | Morgan P Kain Eloise B Skinner Erin A Mordecai |
| Stanford University | | Morgan P Kain |
| Stanford University | Natural Capital Project | Morgan P Kain |
| Stanford University | King Center on Global Development | Erin A Mordecai |

The funders had no role in study design, data collection and interpretation, or the decision to submit the work for publication.

## Author contributions
Morgan P Kain, Conceptualization, Software, Formal analysis, Investigation, Visualization, Methodology, Writing - original draft, Writing - review and editing; Eloise B Skinner, Conceptualization, Data curation, Formal analysis, Investigation, Visualization, Methodology, Writing - original draft, Writing - review and editing; Andrew F van den Hurk, Conceptualization, Data curation, Supervision, Investigation, Writing - review and editing; Hamish McCallum, Supervision, Writing - review and editing; Erin A Mordecai, Conceptualization, Supervision, Funding acquisition, Writing - original draft, Writing - review and editing

## Author ORCIDs
Morgan P Kain ⓘ https://orcid.org/0000-0003-0605-7289
Eloise B Skinner ⓘ https://orcid.org/0000-0002-9032-2710
Andrew F van den Hurk ⓘ https://orcid.org/0000-0001-6262-831X
Hamish McCallum ⓘ https://orcid.org/0000-0002-3493-0412
Erin A Mordecai ⓘ https://orcid.org/0000-0002-4402-5547

## Decision letter and Author response
Decision letter https://doi.org/10.7554/eLife.67018.sa1
Author response https://doi.org/10.7554/eLife.67018.sa2

# Additional files

## Supplementary files
• Source data 1. host_response.csv – Viremic responses of non-human vertebrates from experimental infections. Shown in *Appendix 1—figure 1*; summarized in *Supplementary file 1*.

• Source data 2. human_titre.csv – Viremic response of humans observed during natural infection. Shown in *Appendix 1—figure 1*; summarized in *Supplementary file 1*.

• Source data 3. mosquito_infection.csv – Laboratory infections of mosquitoes: infection probability. Shown in *Appendix 1—figure 3*; summarized in *Supplementary file 2*.

• Source data 4. mosquito_transmission.csv – Laboratory infections of mosquitoes: transmission probability. Shown in *Appendix 1—figure 4*; summarized in *Supplementary file 2*.

• Source data 5. host_abundance.csv – Host densities in Brisbane, Australia. Summarized in *Supplementary file 1*.

• Source data 6. mosquito_feeding.csv – Blood-feeding surveys of mosquito species' found in Brisbane, Australia. Summarized in *Supplementary file 2*.

• Source data 7. host_seroprevalence.csv – Seroprevalnece of vertebrate hosts in Brisbane, Australia. Summarized in *Supplementary file 1*.

• Source data 8. mosquito_abundance.csv – Abundance of mosquito species in Brisbane, Australia. Summarized in *Supplementary file 2*.

• Supplementary file 1. Table S1: Summary of host data – Summarized host titer, seropositivity, and abundance data.

• Supplementary file 2. Table S2: Summary of vector data – Summarized mosquito infection probability, transmission probability, and abundance data.

• Supplementary file 3. Table S3: Summary of mosquito blood meal data — Summarized mosquito blood meal data used in the mosquito feeding preference model.

• Transparent reporting form

## Data availability
All data analyzed and all code generated during this study are included in the manuscript and supporting files.

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

# Appendix 1

## Statistical sub-models

### Vertebrate hosts: titer profiles

We converted reported means and standard deviations for peak titer and duration of detectable titer into continuous titer profiles, which are needed to translate titer into mosquito infection probability given a feeding event. For each species, we first simulated $N$ titer values at each of the first day, the day hosts reached their peak titer, and the last day of infection (where $N$ is the total number of individuals of each species in the infection experiment that developed detectable viremia). We simulated the last day of infection and the log of peak titer for each species by drawing $N$ samples from a Gaussian distribution using the reported means and standard deviations for infection duration and peak titer. We assumed titre on day 1 and the last day of infection were at a detectability threshold of $10^{2.2}$ infectious units/ml blood (the detection limit of RRV in African green monkey kidney (Vero) cells: *McLean et al., 2021*), and that simulated peak titer occurred at the midpoint between the first and simulated last day of infection. We then fit a linear model in R to these simulated data using linear and quadratic terms for day post infection. To quantify uncertainty in quadratic titer profiles, we simulated and fit linear models to 1000 simulated sets of titer curves; in *Appendix 1—figure 1* we show the 95% CI for each of the 15 hosts' quadratic profiles generated from this procedure with the raw summary values of peak and duration of titer extracted from the literature overlayed (the area under the curve for these titer profiles are shown in *Appendix 1—figure 2*).

For human titer profiles, we used data obtained during an epidemic of RRV in the Cook Islands in 1980 (*Rosen et al., 1981*). This study measured human titer from the day of symptom onset; raw data showed that humans experienced peak titer on day 1 of symptoms. To remain consistent with how we modeled non-human titer curves, we fit quadratic curves to the human titer data, which predict a peak at the first day of symptoms and that humans have detectable titer approximately 3 days prior to symptom onset. While it is uncertain how many days prior to symptom onset humans manifest a detectable viremic response, expert opinion on RRV (Leon Hugo and John Mackenzie *pers com*) is that it is likely *at least* 1 day, and for other arboviruses such as dengue, humans produce virus titers sufficient to infect mosquitoes for multiple days prior to symptom onset (*Duong et al., 2015*). Because our assumption of a quadratic titer curve extends titer to 3 days that have no direct quantitative empirical support—which results in humans having a longer duration of titer than any other host—as a conservative estimate of human physiological competence, we also run our model assuming that human titer increases from an undetectable level to a peak on day 1 of symptom onset after only a single day (instead of approximately three as predicted with the quadratic model).

### Mosquito vectors: infection and transmission probability

In total, we gathered data for 17 experimentally infected mosquito species (all extracted data is available as Source Data Files). In these experiments, mosquitoes were fed a given dose of RRV via an artificial blood source which contained diluted stock virus or, in limited cases, from living organisms, such as suckling mice. The proportion that went on to become infected (RRV detected in the body) and infectious (RRV detected in the saliva measured artificially or via feeding on a susceptible vertebrate) was recorded. In the generalized linear mixed effects model (GLMM) for mosquito infection probability, we used virus dose as the sole fixed effect and modeled variation among mosquito species using a random intercept and slope over dose. For transmission probability over time, we used days since infection as the sole fixed effect and modeled variation among mosquito species' transmission over time using a random intercept and slope over time (days since feeding). While the maximum transmission probability is sometimes allowed to vary by mosquito species, we lacked the data to estimate different maxima for each species. Thus, we used simple logistic regression which models probability using an asymptote of one. Uncertainty among mosquito species (which were modeled using a random effect) were obtained from the conditional modes and conditional covariances of the random effect for species (for further details see the code available on GitHub: *Kain, 2021a*).

## Mosquito vectors: feeding behavior

We fit our multinomial model in a Bayesian context because a Bayesian model allows us to incorporate prior probabilities in order to model feeding patterns on species that were either: (A) not detected in the host survey but appear in the blood meal data; or (B) detected in the host survey but do not show up in the blood meal data. Specifically, for case (A), priors allow us to model a mosquito's feeding patterns on a species that would otherwise have an abundance of zero without having to make an arbitrary assumption such as, for example, that a given host species that was not observed in the community but whose blood was observed in a mosquito was exactly equal in rarity to the rarest detected species (e.g. see *Hamer et al., 2009*). For case (B), priors allow us to avoid the biologically implausible assumption that a mosquitoes' preference for a host that simply was not recorded in that specific blood meal survey is exactly zero. For example, in our blood meal data, zero *Culex quinquefasciatus* were recorded to have taken a blood meal from humans, although it is well understood that this species does occasionally bite humans and can lead to human infection of, for example, West Nile virus (*Molaei et al., 2007*). We used a Dirichlet distribution for our prior on host abundance, which is the conjugate prior to the multinomial distribution (*Tu, 2014*). The Dirichlet distribution is parameterized with a vector of positive reals ($\alpha$), with length equal to the number of categories being modeled (for us, hosts). For our Dirichlet prior we smoothed the observed host proportions in the data in an attempt to control for the low detection probability of more cryptic species to produce the following $\alpha$ vector (rounded for display): human = 917, dog = 187, cat 138, bird = 73, possum = 22, flying_fox = 19, cattle = 14, macropod = 7, sheep = 0.4, horse = 0.2, rabbit = 0.2, rat = 0.2.

We assume that the underlying feeding preference of each mosquito species (proportional increases or decreases in biting host species relative to biting those species in proportion to their relative abundance) across host species is Gamma distributed (a flexible two-parameter distribution on [0, inf] that can resemble an exponential distribution with mode at zero or a Gaussian-like distribution with strictly positive values). We allow the shape of this Gamma distribution to vary among mosquito species, which, in biological terms, flexibly allows the model to capture mosquitoes with specialist feeding preferences (skewed Gamma across host species—mosquitoes bite many host species rarely and a few species often) and generalist feeding tendencies (flatter Gamma—mosquitoes bite hosts in accordance with their relative abundance). To do so, we use a multi-level model in which we assume that the shape of the Gamma distributions describing each mosquito species' preference are in turn Gamma distributed. This can be interpreted as being used to model the distribution of specialists and generalists mosquitoes in the sample. Specifically, to allow the 'shape' of the species-level Gamma distributions to vary, we assume that the two parameters that describe those Gamma distributions are drawn from two higher-level Gamma distributions; we used a prior of gamma(4, 4) for each of the higher-level Gamma distributions which are minimally informative priors used to constrain the model to search a realistic space of feeding preferences (e.g. not a perfectly uniform case or an extremely skewed exponential case).

**Appendix 1—table 1.** Reviews suggesting frameworks on how to define the terms 'host' and 'vector' vary greatly in which physiological and ecological criteria they consider (indicated with "X') contribute to the importance of a species as hosts or vectors.

| Reference | Host or vector | Physiological | | | | Ecological | | | | |
| | | Pathogen load (e.g. titre duration and magnitude) | Pathogen detected (e.g. virus isolation) | Immune response (e.g. detectable antibodies) | Survival (i.e. survives long enough to transmit) | Population susceptibility | Abundance | Contact between vector and host | Breeding patterns | Activity patterns |
| --- | --- | --- | --- | --- | --- | --- | --- | --- | --- | --- |
| *DeFoliart et al., 1987* | Host | | X | | | X | X | X | X | |
| *Levin et al., 2002* | Host | X | X | X | | X | | | | |

*Continued on next page*

*Appendix 1—table 1 continued*

| Reference | Host or vector | Physiological | | | | Ecological | | | | |
| | | Pathogen load (e.g. titre duration and magnitude) | Pathogen detected (e.g. virus isolation) | Immune response (e.g. detectable antibodies) | Survival (i.e. survives long enough to transmit) | Population susceptibility | Abundance | Contact between vector and host | Breeding patterns | Activity patterns |
|---|---|---|---|---|---|---|---|---|---|---|
| *Ashford, 1997* | Host | X | | X | | X | | X | | |
| *Haydon et al., 2002* | Host | | | X | | X | X | X | | |
| *Kuno et al., 2017* | Host | X | X | X | | X | | | | |
| *Cleaveland and Dye, 1995* | Host | X | | X | | X | | | | |
| *Silva et al., 2005* | Host | X | | | X | X | | X | | |
| *WHO Scientific Group on Arthropod-Borne and Rodent-Borne Viral Diseases, 1985* | Host | X | | X | X | X | | | X | |
| *Scott, 1988* | Host | X | | X | X | | | X | | |
| *Wilson et al., 2017* | Vector | | | | | | | | | |
| *DeFoliart et al., 1987* | Vector | X | X | | | | | X | | |
| *Kahl et al., 2002* | Vector | X | | | X | | | X | | |
| *Killick-Kendrick, 1990* | Vector | X | X | | | | X | X | | X |
| *Beier, 2002* | Vector | | | | | | | | | |
| *WHO Scientific Group on Arthropod-Borne and Rodent-Borne Viral Diseases, 1985* | Vector | X | X | | | | | X | | |

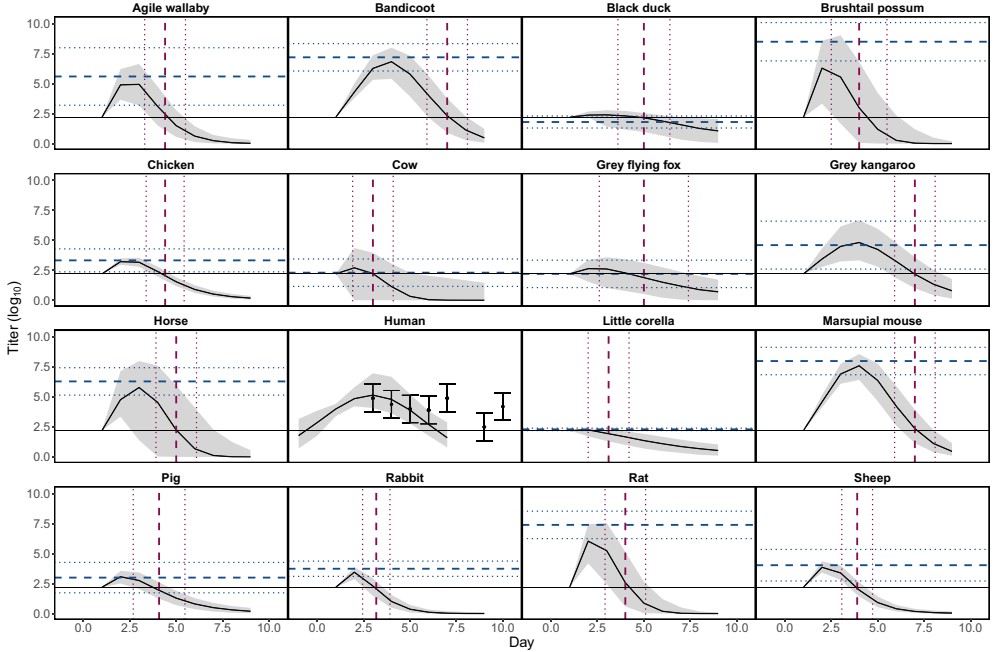

**Appendix 1—figure 1.** Continuous virus titer profiles over hosts' infectious periods constructed using empirical estimates of peak titer and titer duration. For all non-human species 'Day' represents days since experimental exposure to Ross River virus (RRV). Solid black curves and gray envelopes show predicted medians and 95% CI calculated from all simulated titer curves. Horizontal dashed blue lines show empirically estimated peak titers (*Supplementary file 1*) for each species and horizontal dotted blue lines show ± 1 SD. Vertical dashed red lines show empirically estimated end dates of detectable titer and vertical dotted red lines show ± 1 SD. Horizontal solid black lines show the maximum detectable titer. For humans, points show reported means from raw data and error bars show ± 1 SD. The human titer data is shifted in time for visualization purposes (in the raw data the first observation of human titer is recorded on day 1 of symptoms not exposure). Our predictions for humans ignore the outlier data point pictured at day 10, but do simulate titer on days prior to empirically observed titer. For further details see commenting in the R code available on GitHub: *Kain, 2021a*.

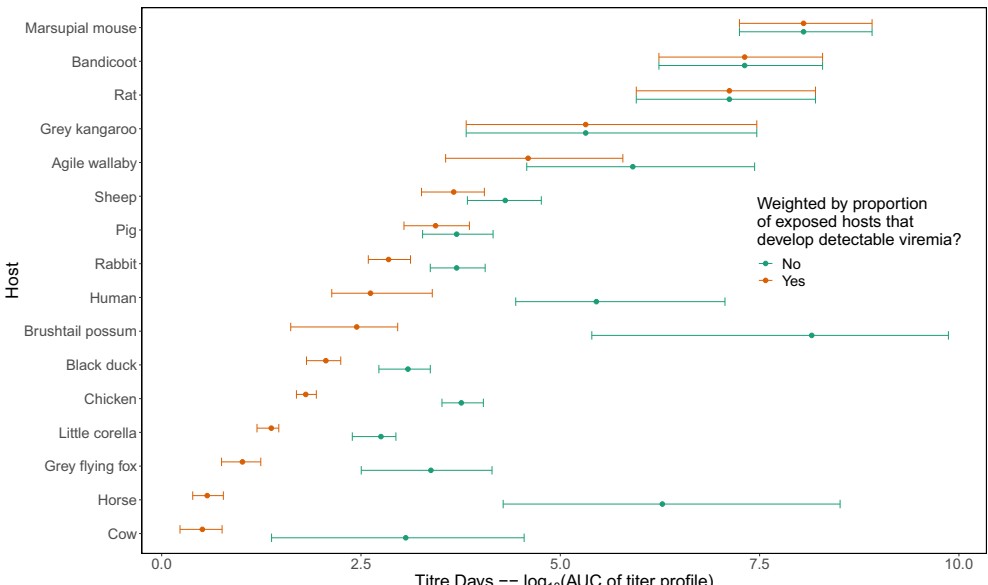

**Appendix 1—figure 2.** Area under the curve (AUC) calculated from the host virus titer curves pictured in *Appendix 1—figure 1*. We use AUC to collapse the continuous host titer curves (*Appendix 1—figure 1*) into a single metric because it simultaneously captures both the height of the curve (actual titer values) and duration of detectable titer (infectious duration). We use AUC to quantify host physiological responses (see *Figure 2A*); however, the complete titer curves (*Appendix 1—figure 1*) are used to host-to-mosquito or mosquito-to-host transmission, not AUC. Orange points and error bars (95% CI) show calculated AUC multiplied by the proportion of all of the individuals of each species that develop detectable viremia when exposed to virus (see *Supplementary file 1* for the proportion of individuals of each species that developed a viremic response in infection experiments). Green points and error bars show calculated AUC ignoring ignoring the proportion of hosts that display a viremic response. Note, for example, the large difference in the physiological competence of horses using these two metrics; horses have been considered important hosts historically, although this claim has ignored the large proportion that do not produce detectable viremia (see *Stephenson et al., 2018*).

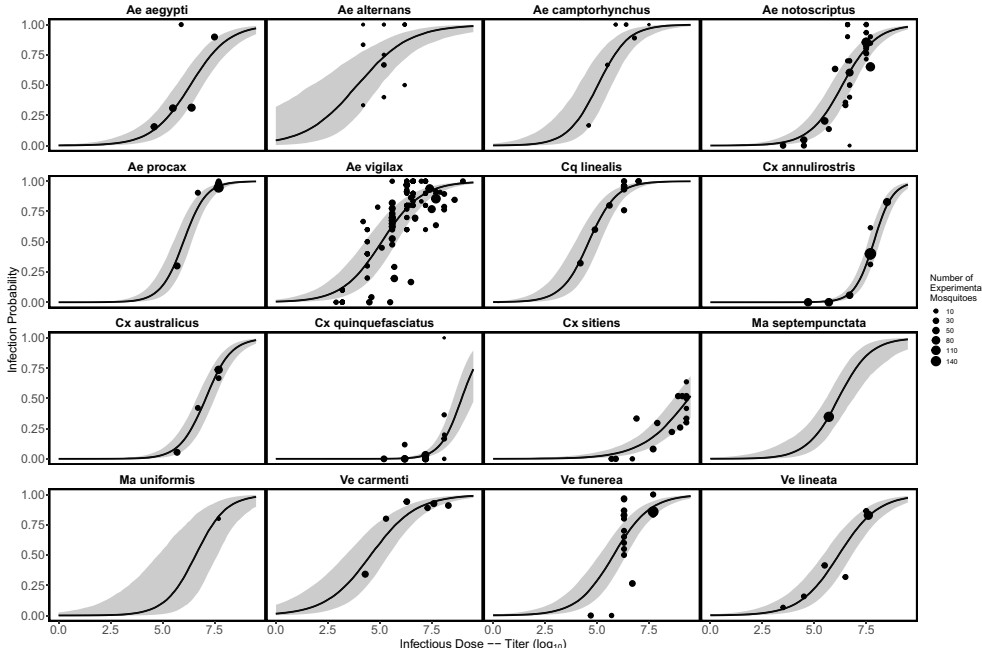

**Appendix 1—figure 3.** Probability mosquitoes become infected with Ross River virus as a function of infectious dose. Data points show the proportion of mosquitoes with infection detected at a given infectious dose in laboratory experiments; point size reflects the total number of mosquitoes exposed to infection. Model predictions are from a binomial GLMM, with dose as a fixed effect and mosquito species as a random effect (intercept and slope over dose), which was fit in R using the package lme4 (*Bates et al., 2015*). Solid black lines show predicted medians, and gray envelopes are 95% CI constructed from the conditional modes and conditional covariances of the random effect (for further details see the code on GitHub: *Kain, 2021a*).

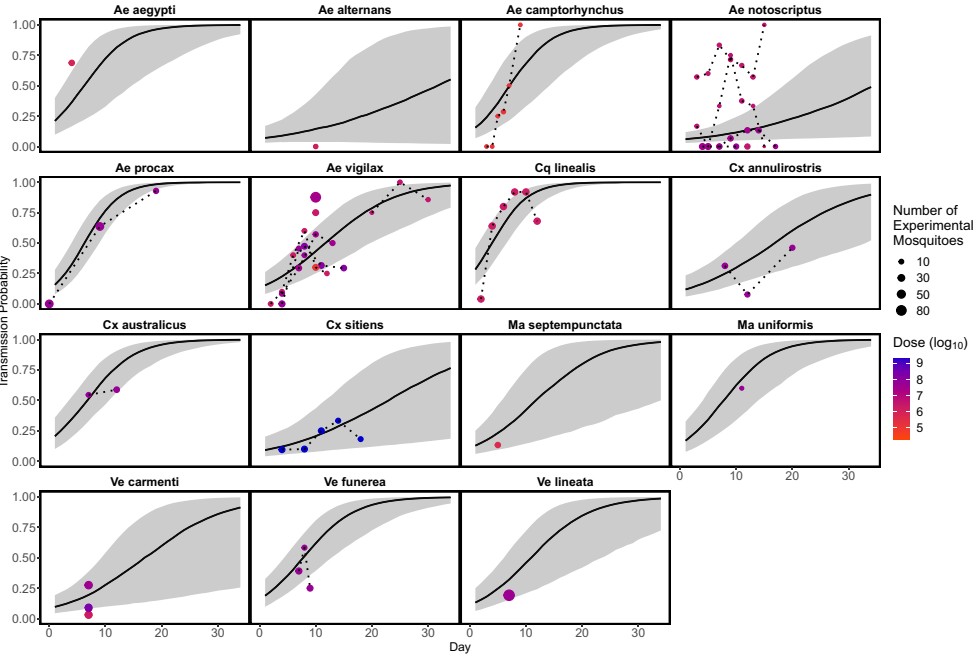

*Appendix 1—figure 4 continued on next page*

*Appendix 1—figure 4 continued*

**Appendix 1—figure 4.** Probability over time that an infected mosquito transmits Ross River virus to a susceptible host given a feeding event. Data points show the proportion of mosquitoes transmitting virus in laboratory experiments ; point size reflects the total number of mosquitoes exposed to infection and color shows the experimental dose mosquitoes were exposed to. Model predictions are from a binomial GLMM, with day as a fixed effect and random effects of mosquito species (intercept and slope over day) and reference (intercept), fit in R using the package lme4 (*Bates et al., 2015*). Solid black lines show predicted medians, and grey envelopes are 95% CI constructed from the conditional modes and conditional covariances of the random effect. We did not include dose as a fixed effect because of model fitting/parameter identifiability issues, but show the doses used in the laboratory experiments here (color). Dotted lines connect data points that are from the same experiment.

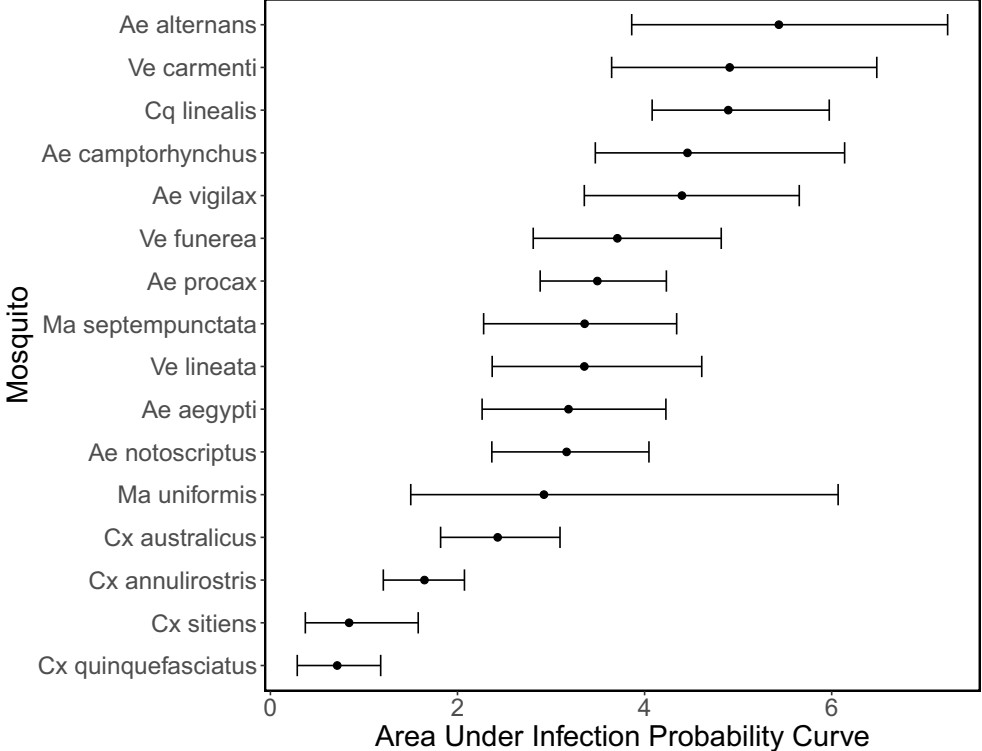

**Appendix 1—figure 5.** Area under the curve of the mosquito infection probability curves shown in *Appendix 1—figure 3*. Points show medians and error bars show 95% CI.

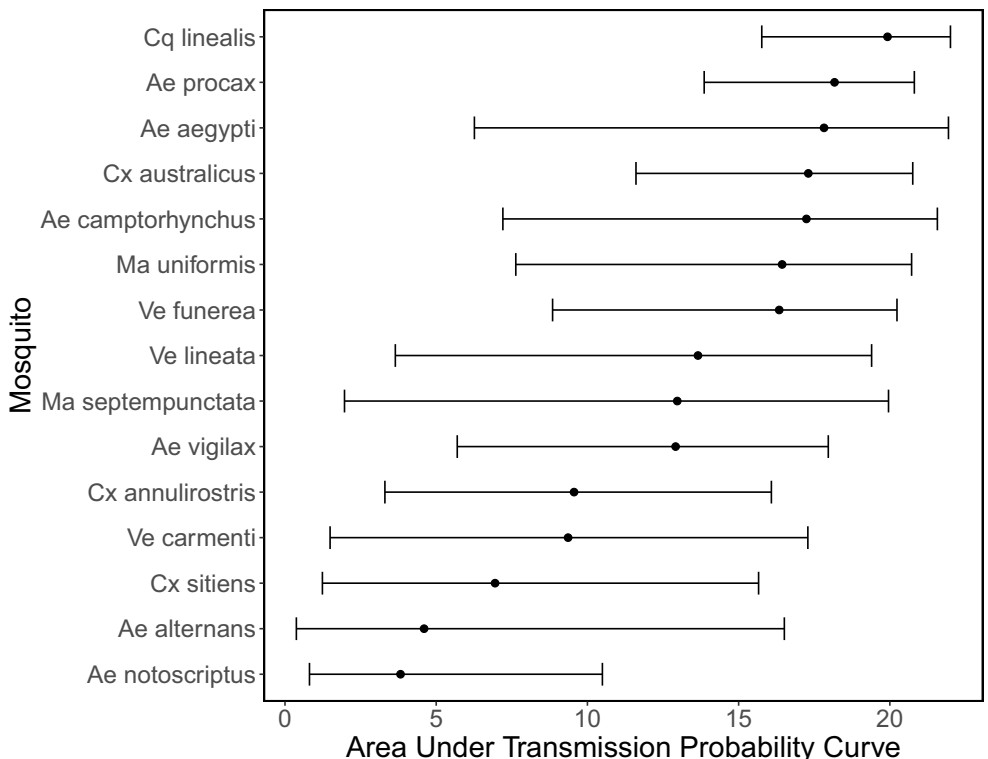

**Appendix 1—figure 6.** Area under the curve of the mosquito transmission probability curves shown in *Appendix 1—figure 4*. Points show medians and error bars show 95% CI. Of all mosquitoes without data just *Ve lineata* is pictured here as in *Appendix 1—figure 4*.

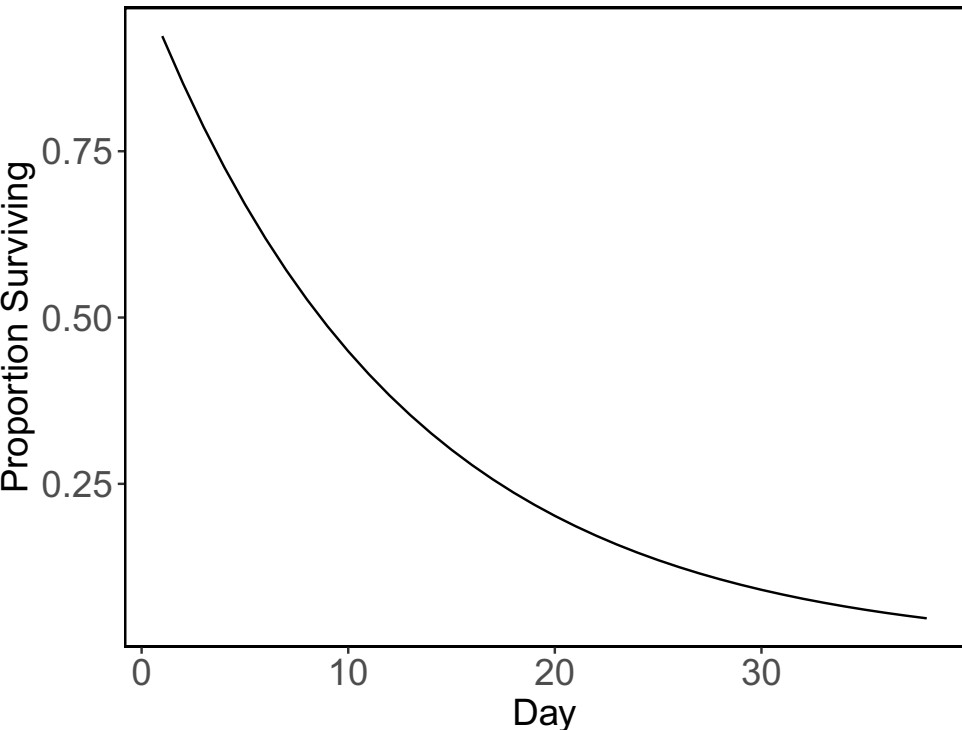

Culex annulirostris survival at half max of optimal laboratory conditions from Shocket et al. 2018 Elife

**Appendix 1—figure 7.** *Culex annulirostris* daily survival in laboratory conditions using the half-max of survival in optimal conditions. In the absence of species-specific survival for most of our species we use this survival curve (from *Shocket et al., 2018* who used data from *McDonald et al., 1980*) for all of the species in our model, but assume that survival after day 38 falls to zero.

# Appendix 2

## Results figures

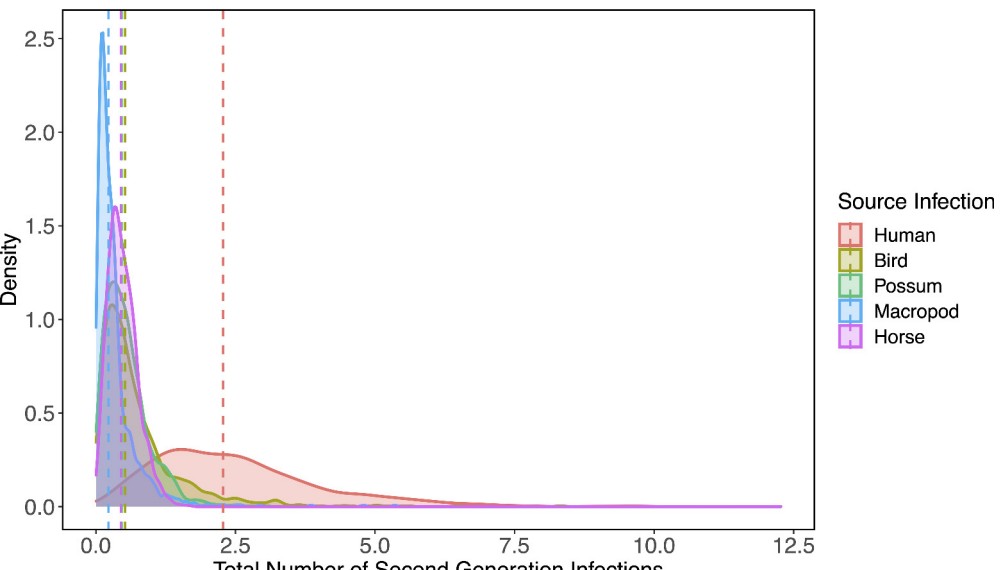

**Appendix 2—figure 1.** Complete density distributions for total estimated host-to-host transmission for the the top five species by median estimates (humans, birds, possums, horses, macropods). Distributions show the 1000 samples obtained by propagating uncertainty from all statistical sub-models see *Table 1* for details. The vertical dotted lines show distribution medians.

## Complete-cycle transmission

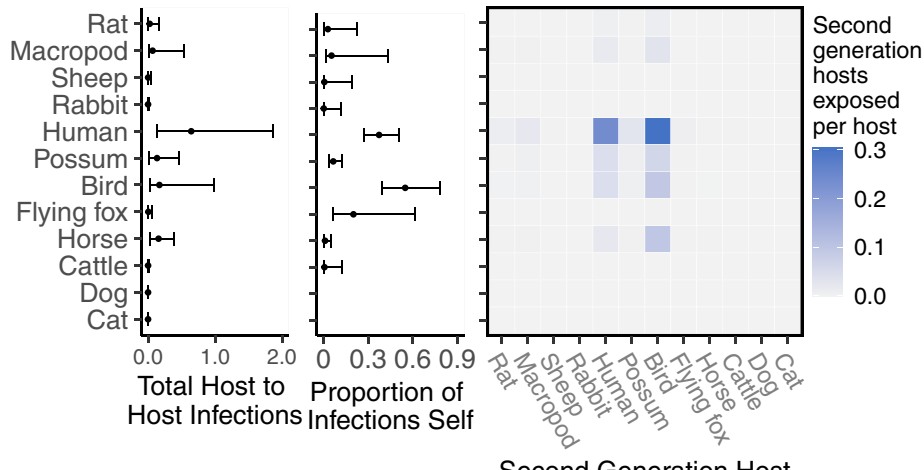

### Second generation "exposures"

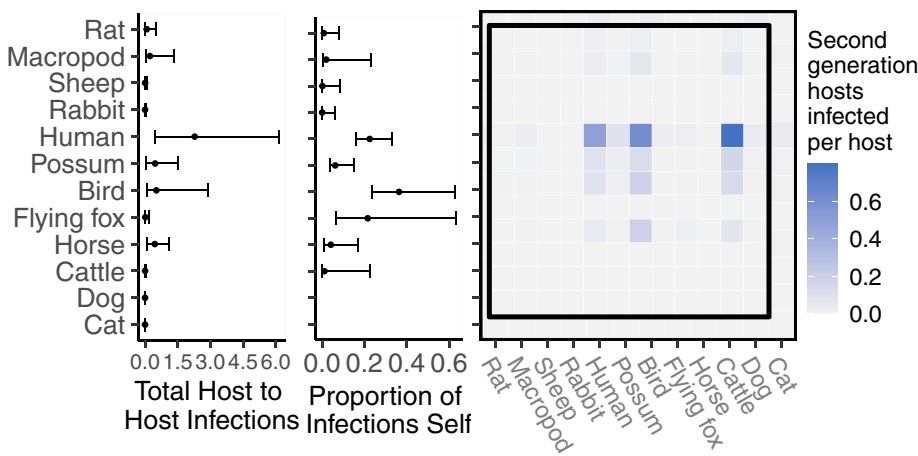

### Second generation "infections"

**Appendix 2—figure 2.** Ross River virus transmission capability of hosts as measured by the number of second generation hosts exposed to infection vs virus transmission capability of hosts as measured by the total number of second-generation hosts that mount a viremic response. The top panel is recreated from *Figure 2C*; the bottom row uses the same calculation for transmission but weights all second generation hosts by the proportion of those hosts that display a viremic response (i.e. dogs do not contribute to the sum in the bottom row). Although host ranks do not change depending on the method of quantifying host transmission importance, overall estimates of transmission decrease when removing sink infections (bottom panel).

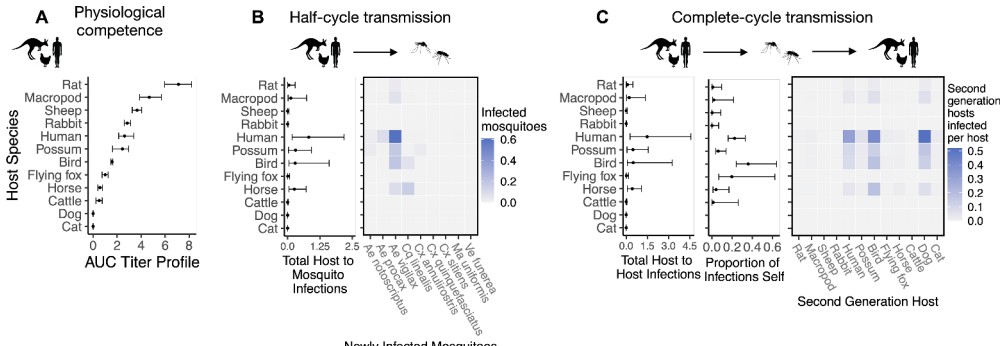

**Appendix 2—figure 3.** Ross River virus transmission capability of hosts based on physiological traits alone or with consideration of ecological traits that drive transmission — assuming human titer begins only 1 day prior to symptom onset instead of assuming a full quadratic titer profile as we do in the main text. A. Physiological response of hosts to experimental infection with RRV, summarized using the area under their estimated titer profiles over time (AUC). In all panels, points show median estimates; error bars are 95% confidence intervals (CIs) that combine the uncertainty from all statistical sub-models used to obtain the estimates presented in that panel (see *Figure 1* and *Box 1* for these components). Titer profile AUC is used only to quantify host physiological competence, while raw titer profiles (pictured in *Appendix 1—figure 1*) are used in half-cycle and complete-cycle transmission. The ordering of hosts based on highest (top) to lowest (bottom) physiological competence in A is conserved in B and C to aid visualization of host order changes among panels. B. Host-to-vector transmission; matrices show the median numbers of vectors infected by each host species, while the points show infection totals (sums across matrix rows), with error bars. C. Host-vector-host transmission. As in B, the matrices show median numbers of next-generation host infections for all host species pairs, while the points show sums across rows of the matrices (left plot) and the proportion of infections in the second generation that are in the same species as the original infected individual (center plot).

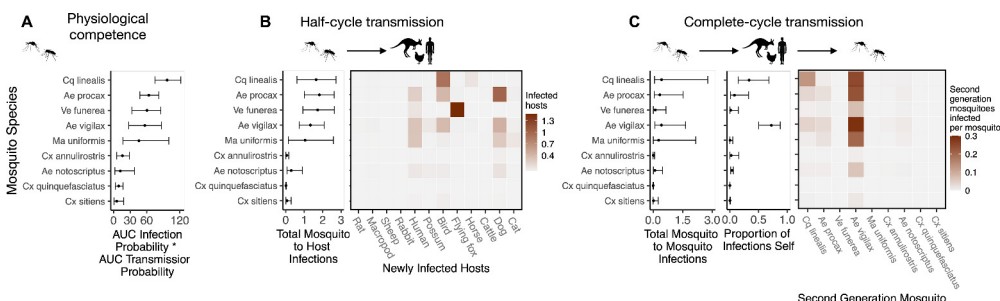

**Appendix 2—figure 4.** Ross River virus transmission capability of mosquitoes based on physiological traits alone or with consideration of ecological traits that drive transmission — assuming human titer begins only 1 day prior to symptom onset instead of assuming a full quadratic titer profile as we do in the main text. A. Physiological response of mosquitoes to experimental infection with RRV, summarized using the area under (AUC) of their estimated infection probability versus dose curves multiplied by the area under their transmission probability versus time curves. Points show median estimates; the error bars in each panel are 95% confidence intervals (CIs) that combine the uncertainty from all statistical sub-models used to obtain the estimates presented in that panel (see *Figure 1* and *Box 1* for these components). AUC is used only to quantify mosquito physiological competence; raw infection and transmission profiles (pictured in *Appendix 1—figure 3* and *Appendix 1—figure 4*, respectively) are used in calculations of half-cycle and complete-cycle transmission. The ordering of vector species based on highest (top) to lowest (bottom) physiological

*Appendix 2—figure 4 continued*

competence in A is conserved in B and C to aid visualization of vector order changes among panels.
B. Vector-to-host transmission; matrices show the median numbers of hosts infected by each vector
species, while the points show infection totals (sums across matrix rows), with error bars. C. Vector-
host-vector transmission. As in B, the matrices show median numbers of next-generation vector
infections for all vector species pairs, while the points show sums across rows of the matrices (left
plot) and the proportion of infections in the second generation that are in the same species as the
original infected individual (center plot).

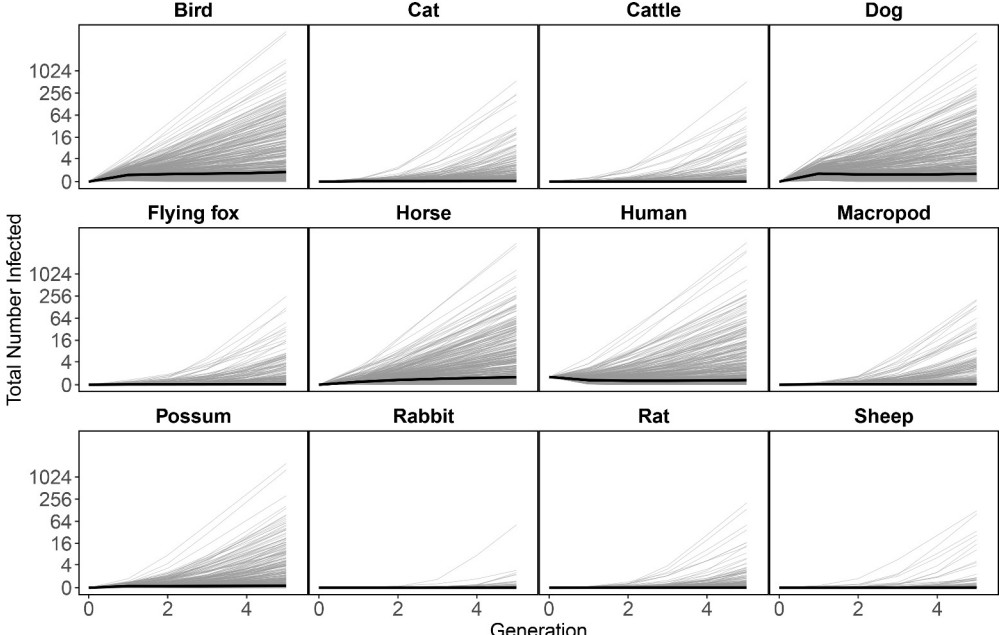

**Appendix 2—figure 5.** An initial human infection propagates infection through the host community.
Starting with a single infected human in generation 'zero' (all hosts begin with zero infected
individuals except humans), the next generation matrix approach can be used to approximate (using
the time step of a generation) how an epidemic would unfold in the community. Here, we show the
total number of new infections of each species as the infection spreads in the community across
generations beginning with the source infection in one human. In generation one, all infections arise
from the source human infection. In subsequent generations, the plotted number of infections for
each species is the estimated total number of infections in that species arising from all transmission
pathways. Our median $\mathcal{R}_0$ estimate for Ross River virus transmission in Brisbane is just above one,
which results in a very slow increase in cases over generations (solid lines); however, large
uncertainty for the number of infections produced by each infected host and mosquito (see
*Figure 2*, *Figure 3*) results in the possibility of explosive epidemics and thousands of infected
individual hosts after a few generations. The thin grey black lines are 500 epidemic realizations.
Because we assume a fully susceptible host and vector population, this is an epidemic simulation,
which would over-estimate the amount of RRV transmission in Brisbane because of the high host
immunity in the host population that is ignored here.

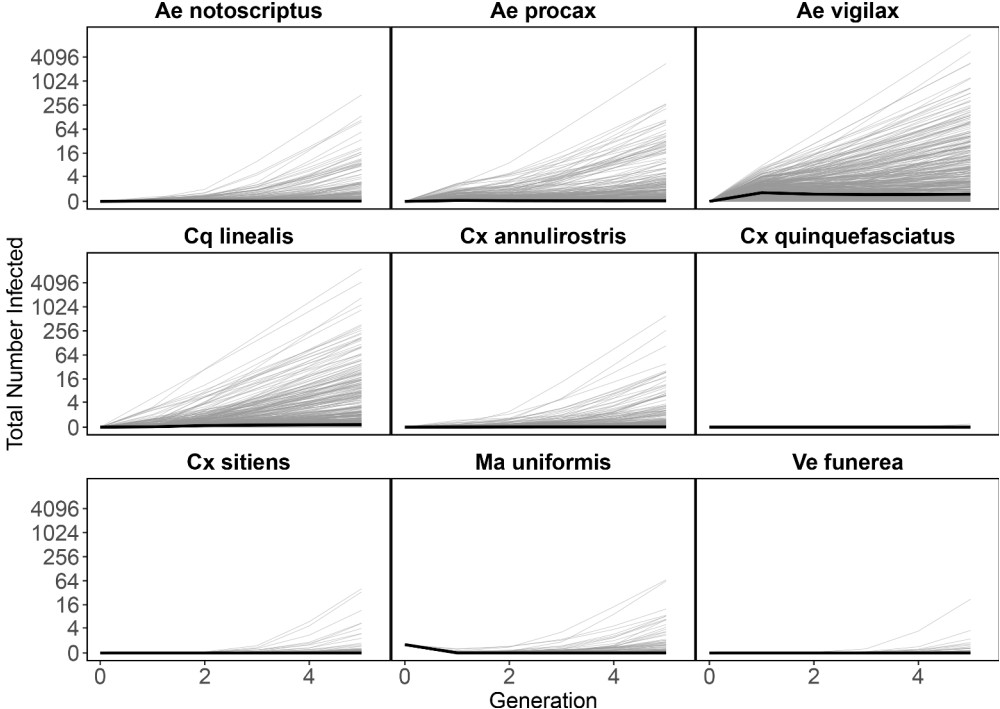

**Appendix 2—figure 6.** An initial *Ma. uniformis* infection propagates through the mosquito community. Starting with a single infected *Mansonia uniformis* in generation 'zero', the next generation matrix approach approximates the number of mosquitoes infected in subsequent generations. All generation one mosquito infections arise from the source *Ma. uniformis* infecting hosts and those hosts infecting mosquitoes; the plotted number of infections for each mosquito species is the estimated total number of infections in that species arising from all transmission pathways. As these results are generated from the same model that produced the results in *Appendix 2—figure 5* (simply with a different perspective) median estimates (bold black line) show slightly increasing numbers of infections in mosquitoes over generations. However, large uncertainty for the number of infections produced by each infected host and mosquito (see *Figure 2*, *Figure 3*) results in the possibility of explosive epidemics and thousands of infected individual mosquitoes after a few generations. As in *Appendix 2—figure 5*, the thin grey black lines are 500 epidemic realizations. Because we assume a fully susceptible host and vector population, this is an epidemic simulation, which would over-estimate the amount of Ross River virus transmission in Brisbane because of the high host immunity in the host population that is ignored here.

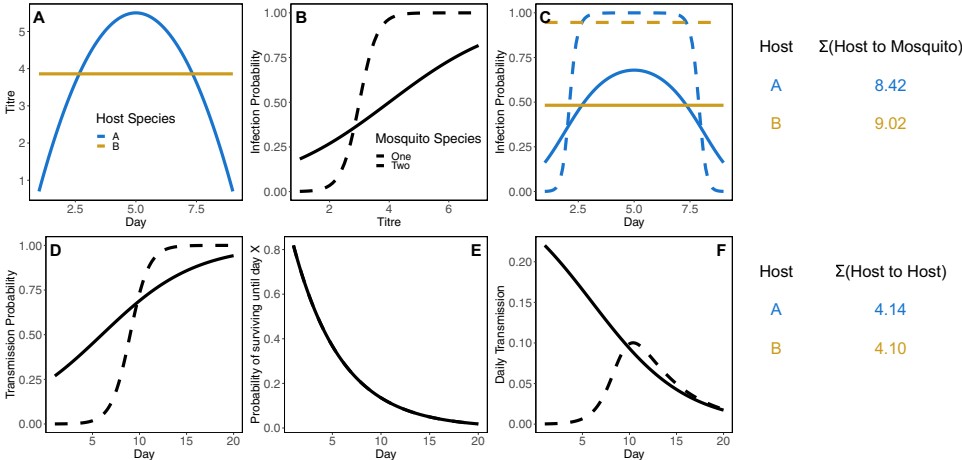

**Appendix 2—figure 7.** Simulated illustrative example for how host species can change rank between host-to-mosquito (panels A-C) and host-to-host (panels D-F) definitions of competence, even without considering host abundance, mosquito abundance, mosquito biting preference, or differences in mosquito survival (each of these variables makes increases the possible routes to host rank reversal). In this example, host species A has a more peaked titer curve than host species B (panel A). Here, when each of these host species are bit by two different mosquito species with different infection probability curves (panel B), host species B has an overall higher probability of infecting these two mosquitoes (panel C). To the right of the top panel shows the total number of mosquitoes infected over the course of 8 days of infection in these two host species, assuming five susceptible mosquitoes of each species per host and a daily biting rate of 0.4 for each mosquito species. When these mosquito species differ in their incubation rate and thus transmission probability (panel D), and the same survival probability (differential survival makes the reversal of ranks easier – if mosquito species two has lower survival the gap between host species will widen) even if they have the same survival probability (panel E), they will have different survival-weighted transmission rates per bite over time (panel F). Taking the total number of infected mosquitoes of each species in the host-to-mosquito infection step and multiplying by the total number of transmissions over the mosquitoes lifetime, considering mosquito biting rate, results in host species A producing a fraction more host-to-host infections than species B.

