## [Decision Letter]

**Acceptance summary:**

This paper nicely highlights the huge amount of data needed to understand the complexities of vector borne disease transmission and control. It produces an elegant framework to rigorously bring together disparate sources of data from multiple hosts and vectors and the results give clear policy relevant results for the control of Ross River Virus in Brisbane.

**Decision letter after peer review:**

Thank you for submitting your article "Physiology and ecology together regulate host and vector importance for Ross River virus and other vector-borne diseases" for consideration by *eLife*. Your article has been reviewed by 2 peer reviewers, and the evaluation has been overseen by a Reviewing Editor and Miles Davenport as the Senior Editor. The following individual involved in review of your submission has agreed to reveal their identity: Gregory Albery (Reviewer #2).

The manuscript represents a considerable body of work and brings together disparate sources of data in a single framework for understanding the relative importance of different vector and host species. This is a welcomed approach to allow the relative contributions (and importantly, the relative uncertainties) to be explicitly investigated.

Essential revisions:

1 – Title should be revised. Though I would agree that physiology and ecology regulate other vector borne diseases this is not shown in this manuscript. The authors should therefore consider removing "and other vector-borne diseases" from the title.

2 – Could the authors explain the rationale for using the ranking framework over something more quantitative such as the use of the basic reproduction number (R0). Ranking systems have the potential to hide considerable nuance, especially when parameterisation is unclear.

3 – The authors mention many of the uncertainties in the discussion, but this is largely ignored in the abstract and results where point estimates which rely on small sample datasets are presented (see comments 1.1 and 1.2 in particular). Given the inevitable lack of good quality data to parameterise the model uncertainty is always likely to be underestimated. Nevertheless, it would be great if the uncertainties could be more fully reflected, including statistical analyses if these were thought to be appropriate. Reviewer 1 highlights that this could be done in a single Bayesian framework. If this is not possible perhaps a figure (along the lines of Figure 1) highlighting the uncertainty of the result would be appropriate. Either way, the abstract and conclusions should be tempered to reflect this.

4 – The model appears to be parameterised for an endemic setting but the multigenerational model highlights invasion dynamics. Why were generations 1,3 and 5 chosen to highlight? The authors state that vector control is conducted in Brisbane to control the disease though this doesn't appear in the parameterisation. Why was this the case and would it not influence the conclusions i.e. it is more likely to target anthropophagic vectors so could reduce their importance.

5 – The paper could be shorted substantially given some of the repetition in the introduction and discussion (a ~10% reduction should be easily possible). Could the authors also considering adding a few sentences of to the results very briefly outlining the methods used in each section (see reviewers comment 2.1).

*Reviewer #1 (Recommendations for the authors):*

The manuscript proposed an interesting approach for an important and famously hard problem. There was much I liked about it, such as approaching the vector-host system in both directions for investigating the complete transmission cycle. I also like the attempt to integrate different sources of data and the results figures were very clear. However, I found the manuscript poorly written (too long, repetitive and not very clear) and there were important gaps in the description of the data that raised concerns about the validity of the methods and strength of the results. I expand on this and provide other suggestions below.

Major concerns (gaps in data and methods description)

1.1 – Methods. It would be important to add statistical support to the results. In particular Figure 2 and 3. Without it the rank results are based on visual observations but with the confidence intervals overlapping so much, they are only very weakly supported. To me, in most cases there is no "winning" host/vector. Humans also have such wide confidence intervals that strong conclusions about it are difficult.

1.2 – Samples sizes. There are no mention of samples sizes. These need to be added for all data. It is particularly important to determine the validity of some results, for example:

– Are the points in Figure Sm3 the data points for vector competence? If so, how can a model be fitted to a single data point and the uncertainty in the predictions is not even wider than for other species with more data? Same seem to occur in Figure Sm5.

– How many bird species and are their competences not variable?

1.3 – Origin of data. Where and how each data type was obtained is never explained. This needs to be clarified for all data. For example, the metric used for physiological competence are titers but how they are obtained?. I assume these are viral titers from titration assays and not PCRs. Are the assays equally sensitive for all species? Are they from vertebrate blood samples and vector saliva? Where do these data come from? Provide references. Perhaps a data section would be helpful.

1.4 – Parameter values. There are multiple assumptions/values that seem taken from literature included in the models that are never described/explained. E.g., infectious period of each host, host abundance, expected average number of infections etc. I appreciate that some data have been previously described, such as vertebrate abundance but a summary here would help understand some of the differences in the results e.g. weighted and non weighted AUCs. Perhaps expansion of Table 1.

1.5 – The use of the term "reservoir" for vector-borne diseases is more confusing than helpful. Who is the reservoir, the vertebrate host or the vector? Maintenance and transmission can't happen without either. Is the complex host-vector the reservoir then? I would suggest avoiding this term throughout for clarity.

1.6 – The first part of the introduction (first 5-6 paragraphs) is mostly redundant and ends up being a repetition of itself and the discussion. Could be reduced to a single paragraph and then straight to the RRV case study, which is an interesting system in itself. The wide applicability of the method is much better explained in the discussion.

1.7 – Figure 1. I didn't find this Figure helpful. On the contrary. It has too many elements that are too small to be visible and understood. Are the graphs from real data or are they make up curves and numbers? I agree that it would be useful to understand where each type of data was used for each step and what model was applied to it. But this Figure is not helping me with that. I would suggest a simplified version, for example in a diagram or schematic without images but that explains the framework, perhaps more like a workflow.

1.8 – Throughout the manuscript the authors mention "the model" to refer to the methodology or framework they develop. This is confusing because there are many models within 'the model' and the model is not actually a model per se… I suggest updating this terminology for clarity.

1.9 – The results seem to fully hang on the definition and estimation of physiological competence. The implications should be well explained.

1.10 – Why was "study" not included as random effect in the models?

1.11 – Why do a single Bayesian model for the mosquito feeding behaviour but not for the other models, and why leave out the rest of the transmission cycle? The approach is piece-meal like with multiple predictions from different GLMMs that are disconnected with one another and their summary statistics are put together at the end. Could these be put into a single Bayesian framework that allows uncertainties and data be propagated throughout the different model components?

1.12 – Bayesian model not defined, what was the likelihood and the priors? Provide reference for priors too.

1.13 – With the size of the human confidence intervals, can we make any statements about the role of human to community transmission?

1.14 – First paragraph of discussion is repetition of introduction.

1.15 – Table S1. Add disease associated with study.

1.16 – Data does not seem available in the GitHub and the model codes need to be organised. Is the Bayesian model missing?

*Reviewer #2 (Recommendations for the authors):*

The work presents a formidable amount of data and I particularly liked the distinction between half cycles and full cycles, and the explicit differentiation between physiological competence and exposure/behavioral processes. The finding that physiological competence may not align with epidemiological salience is an important one for vector-borne pathogens in general, and a useful lesson going forward. The paper falls short a bit on its description of the modelling process in relating them to the results, and the novelty of the finding that physiological competence does not fully explain epidemiological importance needs to be clarified, but overall this is a solid and useful contribution. The following points should be addressed

2.1 If the paper is going to be in results-first format, the results need a bit more description of the models to facilitate understanding. I found myself wondering how exactly these results were being produced, and needing to flit back and forth from the results to the methods, which isn't ideal. Including the right level of detail (i.e. without recapitulating the methods) is tricky in a results-first Results section, but I think this paper could definitely do with including more information. A few sentences at the start of each section giving a bit of background on the model formulation and how the answers were produced would do the job.

2.2 The models are doing a very heavy lift here (particularly the NGM's), and I appreciate the authors' decision to include model limitations fairly prominently in their discussion. I have no personal experience with models like these, but they appear to have been conducted well.

2.3 Although the authors' reference to host species' ecology as an important determinant of their epidemiological role is useful, the discussion particularly could do with zeroing in more precisely on what aspects of the hosts' ecology are driving the disparity between physiological competence and epidemiological importance (i.e., behaviors, vector biting preferences, and population dynamics). The findings in the paper are definitely novel, but without clarifying what might be going on to drive this disconnect there's a risk of reinventing the wheel of behavioral competence a bit. These processes are all decomposed in Cynthia Downs and colleagues' 2019 competence paper in Trends in Parasitology (for example), and I think the paper should be a bit more explicit in the introduction and discussion that there are well-appreciated reasons that we might not expect physiology to paint the whole picture of competence. For example, it is relatively unsurprising that very rare hosts are epidemiologically unimportant in the system irrespective of their physiological competence. Is it possible that humans' role in these dynamics could be reduced to "there are lots of humans, so they end up being important"? The fact that physiological competence is only one component is in the introduction (lines 54-58), but a more detailed outline of the traits (behavioral and demographic) that could override it is necessary. This will then set the stage for the findings of the models.

[Editors' note: further revisions were suggested prior to acceptance, as described below.]

Thank you for submitting your article "Physiology and ecology combine to determine host and vector importance for Ross River virus" for consideration by *eLife*. Your article has been reviewed by 1 peer reviewer, and the evaluation has been overseen by a Reviewing Editor and Miles Davenport as the Senior Editor. The reviewers have opted to remain anonymous.

Essential revisions:

The revised manuscript is much clearer and nicely highlights the huge amount of data needed to understand the complexities of vector borne diseases. We feel that the article is very close to being acceptable for publication but would like to clarity on points 4 and 5 highlighted by Reviewer #1 below before we proceed. This would allow clarity on the fitting process and would improve interpretation of the titre data (no changes needed to model, just discussion). When preparing the final manuscript, the authors are encouraged to consider the other points highlighted below by Reviewer #1 which could provide further clarification to the reader.

*Reviewer #1 (Recommendations for the authors):*

The authors seem to have put a lot of thought in the revised manuscript and have addressed most of my concerns well. Thank you. However, this is a very dense manuscript and I still find the methods confusing in places.

1 – Figure 1 is much more informative now but perhaps could be expanded to also provide an indication of the type of model used in each parameter to show the workflow.

2 – On previous comment 1.1. I appreciate the revisions and the more honest presentation of the results. All much clearer. I also welcome the explanation of propagation of uncertainty, this is an important feature that I have missed. One of my key points here though was that regardless of how uncertainty is estimated, a formal quantitative way of comparing the densities estimated (such as z-test, Wilcoxon or Kolmogorov tests?) could strengthen the outcomes.

3 – On previous comment 1.11. I don't agree with the reasons mentioned for not using a single Bayesian model instead of the piece-meal approach – 2/3 are considered advantages of Bayesian modelling and the less-friendly approach is debatable (indeed the framework proposed in this manuscript does not seem very user-friendly for empiricists either). That said, although I would approach it differently, I cannot say it is incorrect – though I raise here the concern about how Bayesian and ML-type outcomes are combined in the NGM. The outcomes of a Bayesian model are fundamentally different from those so-called frequentist approaches. All your parameters are 'frequentist' apart from feeding behaviour. Are these being incorporated in a comparable way?

4 – Titre profiles. I have two standing concerns with these. Not sure if problems or if they just need to be explained and discussed.

4.1. Titration values tend to be highly variable within species, let alone between species. Given the small differences in the results, how confident can we be in that the results are meaningful at the ecological scale used?

4.2. The analysis are based on peak titre but considering the different profiles this could be problematic. E.g. in the example shown (Appendix 2 Figure 7: A) where one species has a marked high peak and the other has a constant titre lower than the peak of the other species. How can we link this to an infection/transmission probability? Especially the short time frame of the experiments.

5 – Appendix 1 Figure 4: Still don't understand how these curves are generated. I appreciate that a baseline curve is being generated from all the data combined (all studies and species). But then how can you differentiate among species. A species random effect is added but, on most occasions, there is 1 transmission value for 1 time point for a single species. Did the model pass validation diagnostics? There is also some confusion with the use of dose: the legend first mentions that dose was used as a fixed effect and then a few lines down says it couldn't be used as fixed effect. This also raises issue about the meaning of the transmission probability, especially post ~15 days as all is driven by a single species (no data for all others).

6 – Appendix 1 Table 1: swap word 'reservoir' for 'host' to be consistent with main text.

7 – Figure 2 add sample sizes from each species.

– L98-100. cattle, horses and flying foxes seem all pretty similar to me

---

## [Author Response]

Essential revisions:1 – Title should be revised. Though I would agree that physiology and ecology regulate other vector borne diseases this is not shown in this manuscript. The authors should therefore consider removing "and other vector-borne diseases" from the title.

We have removed the "and other vector-borne diseases" phrase as suggested.

2 – Could the authors explain the rationale for using the ranking framework over something more quantitative such as the use of the basic reproduction number (R0). Ranking systems have the potential to hide considerable nuance, especially when parameterisation is unclear.

We now clarify that we used a quantitative R0 framework (the Next Generation Framework [NGM]); results now focus on interpreting the continuous values estimated by this quantitative framework and their confidence intervals (which Figures 2 and 3 present). The contribution that each host and mosquito species makes to transmission as calculated in the NGM model is detailed on lines 481-492 and lines 529-544.

– We can see that our previous over-emphasis on species ranks has generated confusion regarding what our model actually calculates. We have revised this language to make it clearer that our framework produces continuous estimates of species competence, and that we discuss grouping of species because the confidence intervals on species quantitative competence values overlap substantially with some species but are distinct from others. To clarify that our framework estimates continuous, quantitative measures of transmission potential (e.g., the number of hosts infected per host) and not ranks, we have:

A) Rephrased goal #1 in the Introduction (removing the term "rank" in favor of species' relative importance)

B) Replaced explicit statements about species ranks with more descriptive statements using the outcomes that the model actually calculates (e.g., the number of mosquitoes infected per host or the number of hosts infected per mosquito). For example, see lines lines 98-100; 126; 129-131; 172-174. While we retain the use of the term "rank" in a few instances in the Discussion as a means of distilling some of the results, we have removed "rank" entirely from the Introduction, Methods, and Results.

C) Explicitly stated what our model estimates (and thus what we show in the Results) in a series of highlight/recap sentences at the start of each Results section that state the most important points of the Methods that are needed for a reader to interpret that specific Results section given the results-first format of eLife (a change suggested by Reviewer 2).

3 – The authors mention many of the uncertainties in the discussion, but this is largely ignored in the abstract and results where point estimates which rely on small sample datasets are presented (see comments 1.1 and 1.2 in particular). Given the inevitable lack of good quality data to parameterise the model uncertainty is always likely to be underestimated. Nevertheless, it would be great if the uncertainties could be more fully reflected, including statistical analyses if these were thought to be appropriate. Reviewer 1 highlights that this could be done in a single Bayesian framework. If this is not possible perhaps a figure (along the lines of Figure 1) highlighting the uncertainty of the result would be appropriate. Either way, the abstract and conclusions should be tempered to reflect this.

This is an important point. We have worked to improve our description of how uncertainty is incorporated into the model throughout the manuscript. Specifically, our framework propagates all uncertainty from all fitted statistical sub-models into the final transmission calculations. We now state more clearly in the Methods which parameters are derived from statistical sub-models (which propagate uncertainty) versus point estimates, raw data, or assumptions (which do not propagate uncertainty) (see lines 546-560). In this section we describe the three ways in which uncertainty from host titer curves, mosquito infection probabilities, mosquito transmission probabilities, and mosquito biting preferences are incorporated into all measures of competence in which these parameters appear. It is true that we used a few point estimates and in some cases raw data instead of statistical model predictions, mainly in cases where the data were too limited to formally fit a statistical model. To clarify these points we have:

A) Have added an "uncertainty" column to Table 1 that describes whether and how uncertainty was propagated.

B) Explicitly state in Results Figures 2 and 3 what uncertainty is contained within the CI.

C) Simplified Figure 1 and included the source for each component involved in transmission (e.g., model predictions or raw data) and mention which components have uncertainty propagated.

D) Added a new 'Box 1' in the Introduction (which synthesizes the three nested components of our framework; we give more information on 'Box 1' in our response to 5 below).

E) Included sample sizes in all Appendix Figures and Supplemental Tables.

F) Reduced the strength of our inference about human importance in the Discussion and Abstract.

– We note that our strategy achieves a similar effect to a full Bayesian model (more detail on why we chose to fit sub-models separately instead of a full Bayesian model in our response to Comment 1.11).

– Finally, we note that despite overlapping 95% CI between humans and the other top species (birds, horses, and macropods), much of the density of human host-to-host transmission potential resides above the upper bound of the 95% CI of all of the other species (specifically, 32% of the human density distribution lies above the upper 95% CI of birds, the next-highest species). This indicates that despite overlap in the predicted density functions of transmission potential, we are confident that humans are at least among the most competent, and potentially the single most competent, host species we studied. We point this out in the Results on line 153 and cite a new Appendix Results Figure (now Appendix 2-Figure 1) which we hope will also help to convey the uncertainty propagation (by showing the full density distribution of the outcome instead of just a CI).

4 – The model appears to be parameterised for an endemic setting but the multigenerational model highlights invasion dynamics. Why were generations 1,3 and 5 chosen to highlight? The authors state that vector control is conducted in Brisbane to control the disease though this doesn't appear in the parameterisation. Why was this the case and would it not influence the conclusions i.e. it is more likely to target anthropophagic vectors so could reduce their importance.

We can see how these decisions generated some confusion. In addition to considering the role of each host and vector species in the endemic setting (Figures 2, 3), we also wanted to consider their role in an epidemic setting (Figure 4) to highlight those hosts and mosquitoes that would play the largest role in disease emergence in a naive community. We focused on an emergence in a new community given evidence that RRV distribution is expanding. We appreciate that our focus on Brisbane, an endemic community, is not ideal, but this was the only location with sufficient data available to parameterize the model. Further, while it is true that at no time or location would all hosts be susceptible to infection in Brisbane, spatial and temporal heterogeneity in Brisbane will lead to different transmission scenarios than the specific endemic setting we consider for the main analyses. Such variation often leads endemic diseases to be analyzed with R0 to provide a baseline metric of transmission potential, which is similar to what we provide here. We have revised the language in the Methods section "Multi-generation approximation" (lines 645-670) to clarify these points. We have also added a recap sentence at the start of the Results section "Multiple generations of transmission” (lines 191-194) to remind the reader why we made this shift (or to first explain it given the Results-first structure).

– Your point about generations 1, 3, and 5 is a good one. We have revised Figure 4 to show generations 1-3 sequentially, rather than generations 1, 3, and 5, and we state in the Results (lines 200-204) and in the Figure 4 caption that pairwise transmissions have converged by generation 3 (which is why we don’t show generations 4 and 5 in the main text). We also now describe our rationale for five generations (which we show in what is now Appendix 2-Figure 5 and Appendix 2-Figure 6) on lines 661-662 of the Methods and restate it on line 280 of the Discussion.

– Though vector control does occur in Brisbane, it is often reactive and sporadic. We do not have data for these sporadic efforts and thus do not include it in the framework. We also note that we do not model seasonal transmission (with, for example, fluctuations in the abundance of hosts and mosquitoes); see lines 295-301 for an expanded discussion about seasonality. While we recognize that RRV transmission will vary by season (lines 295-301), we believe our data synthesis and model framework is novel for RRV transmission within Brisbane even without including seasonality or vector control. We have added details stating that the host and mosquito abundance surveys were conducted between October and May (lines 580-581) to help provide context for which season the presented predictions are best suited to represent.

5 – The paper could be shorted substantially given some of the repetition in the introduction and discussion (a ~10% reduction should be easily possible). Could the authors also considering adding a few sentences of to the results very briefly outlining the methods used in each section (see reviewers comment 2.1).

We have streamlined the Introduction to remove the redundancies highlighted by Reviewer 1. Also, to help streamline the Introduction (and to assist in the Results-first format) we have moved the details about the specific steps of the framework (physiology, half-cycle and complete-cycle transmission) into a new 'Box 1'. We tie this to Figure 1 to help link the life-cycle of RRV, the workflow of the model, and the details about the three ways of estimating competence. We hope this will both help the flow of the Introduction as well as assist with the Results-first format (re: Reviewer 2).

– We have reduced the length of the Discussion by removing similar restatements about physiology vs. ecology and host and vector importance.

– We have also followed the excellent suggestion to add a few sentences to the start of each Results section to assist readers given the Results-first format (re: Reviewer 2).

– The combined Introduction and Discussion are now ~4,500 words (including Box 1), down from ~5,650 words in our original submission. However, we note that despite cutting over 1,000 words from the Introduction and Discussion, the new text in the Methods and in the Results to help clarify our framework has resulted in our revision having an almost identical word count to the original submission (just about 10,200).

Reviewer #1 (Recommendations for the authors):The manuscript proposed an interesting approach for an important and famously hard problem. There was much I liked about it, such as approaching the vector-host system in both directions for investigating the complete transmission cycle. I also like the attempt to integrate different sources of data and the results figures were very clear. However, I found the manuscript poorly written (too long, repetitive and not very clear) and there were important gaps in the description of the data that raised concerns about the validity of the methods and strength of the results. I expand on this and provide other suggestions below.Major concerns (gaps in data and methods description)1.1 Methods. It would be important to add statistical support to the results. In particular Figure 2 and 3. Without it the rank results are based on visual observations but with the confidence intervals overlapping so much, they are only very weakly supported. To me, in most cases there is no "winning" host/vector. Humans also have such wide confidence intervals that strong conclusions about it are difficult.

This is an important point. We can see how some of our previous language in the Results and Discussion may have inadvertently oversimplified the results and glossed over the degree of overlap in confidence intervals. We have carefully revised the text to make sure that all references to human and bird importance are stated with a caveat about large uncertainty or accompanied with words such as ("potential" or "may"). For example, see lines 248 and 408 in the Discussion or lines 96-98 and 120-122 in the Results. In the Abstract we have removed explicit reference to humans in favor of sticking to a broader statement about the species with the highest physiological competence were not the most important for community transmission (left unsaid but for rats and sheep 95% CI don’t overlap with humans for for host-to-host).

– We can see that our previous version did not make it clear enough that all of the uncertainty in all sub-models fit to data was propagated through to the estimates of half-cycle and full-cycle transmission. We now clarify how uncertainty is propagated from fitted sub-models (lines 545-560) and list the sources of data and parameters that do not have uncertainty in Table 1. In the captions of Figures 2 and 3 we also now state what uncertainty is propagated. We also briefly state in the Results that uncertainty is propagated from all statistical sub-models (lines 146, 152). Finally, in our revamped Figure 1, we list which components of the framework have uncertainty propagated. We address your other points about data and sample sizes in our responses to your other comments below.

– We also now more clearly emphasize the point that despite overlapping 95% CI between humans and the other top species (birds, horses, and macropods), much of the density of human host-to-host transmission potential resides above the upper 97.5% confidence bound of all of the other species (specifically, 32% of the human density distribution lies above the upper 95% CI of birds, the next-highest species; Results on lines 122-126). We have added Appendix 2-Figure 1 to further emphasize the point that we propagated all uncertainty from statistical sub-models (more on uncertainty propagation in our comments below).

– Finally, given that the estimates we present in the Figures are quantitative measures of transmission (the number of infected mosquitoes per infected host) and not ranks, we have removed the term "rank" from the text (except for a few intentional uses in the Discussion) to avoid confusion about what our model estimated.

1.2 Samples sizes. There are no mention of samples sizes. These need to be added for all data. It is particularly important to determine the validity of some results, for example:– Are the points in Figure Sm3 the data points for vector competence? If so, how can a model be fitted to a single data point and the uncertainty in the predictions is not even wider than for other species with more data? Same seem to occur in Figure Sm5.– How many bird species and are their competences not variable?

Good point. We have added sample sizes to all Supplemental Figures (now Appendix Figures) and Tables (now attached as Supplementary files: Tables S1-S3). The points in Figure Sm3 (now Appendix 1- Figure 3) are raw data from individual infection experiments (the proportion of mosquitoes found infected given a specific dose). We have expanded the caption of Appendix 1-Figure 3 to clarify this, and we have now sized the points by the number of mosquitoes exposed in each of those infection experiments.

– The confidence intervals in Appendix 1-Figure 3 and Sm5 (now Appendix 1-Figure 4) are generated directly from the logistic regression mixed model, which estimates the infection probability versus dose relationship for each mosquito species while, effectively, borrowing information on the more-studied species to inform the curves for the less-studied species. The confidence intervals are wider for species with less data (e.g., Ma. uniformis and Ma. septempunctata), which is displayed in Appendix 1-Figure 5 (they have the widest 95% CIs) as the area under the curves pictured in Appendix 1-Figure 3 (considering the uncertainty in those curves). Similarly, for the mosquitoes in Appendix 1-Figure 4, the species with fewer data points have wider confidence intervals, which can be seen in Appendix 1-Figure 6. However, it is true that they are only a little wider. Because of the information borrowing in the mixed model, the estimates for these species (that have very little data) are strongly pulled to the overall "grand" mean, which is estimated with a reasonable amount of certainty given the overall number of data points and the number of species with relatively similar responses. Also, we note that an individual species with even one data point that falls very near that mean response (e.g., Ma. septempunctata, which had been a large sample size) will produce narrower 95% CIs than will be produced for a species with one data point away from this grand mean with a smaller sample size (e.g., Ma. uniformis which has the largest 95% CIs). As a specific example: given that so few species have means that rise to, for example, an infection probability of 50% at a low dose, the CI for even mosquitoes with no data around these doses range will not include an infection probability of 50%. This uncertainty for less studied mosquitoes is propagated into the calculations for transmission, which can be seen in the very wide CI for Ma uniformis (all panels Figure 3).

– We describe our simplification of the bird's responses into "bird" on lines 626-633. We state there that this is a necessary simplification to run the model given how few birds have been infected. We now point out that the physiological response of the three birds are clustered together in Appendix 1-Figure 2 (Methods lines 626-633 and Discussion (Caveats section) lines 341-342), which provides some support for this simplification.

1.3 Origin of data. Where and how each data type was obtained is never explained. This needs to be clarified for all data. For example, the metric used for physiological competence are titers but how they are obtained?. I assume these are viral titers from titration assays and not PCRs. Are the assays equally sensitive for all species? Are they from vertebrate blood samples and vector saliva? Where do these data come from? Provide references. Perhaps a data section would be helpful.

We can see that some of our explanations of data were not clear enough. We have now:

A) Provide additional details about the data (and the data collection) used in each component of the framework in Methods sub-section that describes that framework component (the largest amount of new material being added to explain the host and mosquito abundance data; see lines 561-572 and 573-592 respectively).

B) Made sure to cite all papers from which we gathered data in these sections.

C) Expanded the details about the data and citations in what are now Tables S1-S3 (attached as Supplementary files), which should also help with this.

D) Added a Methods section titled "Mosquito abundance" (lines 573-592) which was missing in our first submission.

– All of the raw data have now been uploaded to GitHub. Our mistake of not uploading those data until now.

– We recognize that the previous submission was light on the details about the experimental methods in the publications from which we extracted data and acknowledge the influence that different experimental assays and methods can have on results. We feel, however, that reporting these details and summarizing the variation among studies (n = 22 if vector and host studies are considered) to a high level of detail would add relatively little value (and some clutter) to an already long manuscript, especially because these details can be found in one place (reviewed in detail in Stephenson et al., 2018). As such, we have now added a reference to Stephenson et al., 2018 which reviews these experimental methods for RRV hosts. We have also added the infectious units each study reports in the reworked supplemental data tables which provides at least some indication to the assay conducted.

1.4 Parameter values. There are multiple assumptions/values that seem taken from literature included in the models that are never described/explained. E.g., infectious period of each host, host abundance, expected average number of infections etc. I appreciate that some data have been previously described, such as vertebrate abundance but a summary here would help understand some of the differences in the results e.g. weighted and non weighted AUCs. Perhaps expansion of Table 1.

Thank you for pointing out these omissions. We now state the exact values that we assumed for mosquito-to-host ratio and how this relates to mosquito abundance on lines 585-592 and have added the assumed point estimate value for mosquito biting rate (0.5/day) in Table 1.

– We can also see that some of our descriptions of model parameters and assumptions were insufficient. We now specifically define host "infectious period" in the equation for host-to-mosquito transmission (Equation. 1) and lines 496-497 and line 650 of the Methods. For mosquitoes, we now state in Equation. 2 and lines 514, 600, and 651 of the Methods that we assume that no mosquitoes survive beyond day 38, and thus we model transmission of all mosquitoes only until day 38 (weighted by an exponentially decaying survival). In continuation of our overall effort to more directly describe what we modeled, we have removed most of our uses of "infectious period."

– We have also reorganized the Methods sections "Vertebrate hosts: titer profiles” (starting on line 425) and "Mosquito vectors: infection and transmission probability” (starting on line 450) so that they first state that we modeled continuous functions over time (or dose), and second describe how we summarize those continuous curves using AUC as a means of collapsing these continuous curves into a metric to compare among hosts and mosquitoes. In these sections we also point out that AUC is used only as a metric to describe these continuous curves, and that it is not used to calculate metrics of transmission (e.g., host-to-mosquito transmission) (see our comment below and in response to Comment 1.9 as well).

– We have rewritten the caption for Appendix 1-Figure 2 to more clearly state how the weighted and non-weighted AUC differ (AUC conditional on the proportion of hosts showing a viremic response or not, respectively). In the caption we also note that the weighted vs. non-weighted comparison is primarily for visualization purposes to highlight how large of a difference considering the proportion of hosts that display a viremic response makes to AUC (in the text we point out that horse titer curves have been viewed historically without conditioning on the fact that few horses actually produce detectable viremia). Finally, we also added a statement in the caption about the use of AUC to summarize the continuous curves to compare hosts (more on this in our response to Comment 1.9) and not in transmission calculations.

– The expected average number of infections is a modeling outcome (for example, the number of mosquito infections per host infection, which is pictured in Figure 2B, which is dependent on mosquito abundance, biting preferences, host titer). Hopefully our refined language around AUC, removal of the term "rank", and our response to your Comment 1.8 that we use "model" to refer to different levels of analysis (sub-models and the overall framework) will resolve this confusion.

1.5 – The use of the term "reservoir" for vector-borne diseases is more confusing than helpful. Who is the reservoir, the vertebrate host or the vector? Maintenance and transmission can't happen without either. Is the complex host-vector the reservoir then? I would suggest avoiding this term throughout for clarity.

We appreciate that infected hosts without mosquitoes would have little importance. To reduce confusion we have removed the term "reservoir" and just use "host."

1.6 – The first part of the introduction (first 5-6 paragraphs) is mostly redundant and ends up being a repetition of itself and the discussion. Could be reduced to a single paragraph and then straight to the RRV case study, which is an interesting system in itself. The wide applicability of the method is much better explained in the discussion.

Following this suggestion, we have streamlined the Introduction down from 8 to 5 paragraphs (and moved the details about the three nested steps of competence into Box 1) and left most of the discussion of the broader applicability of the method to the Discussion.

1.7 – Figure 1. I didn't find this Figure helpful. On the contrary. It has too many elements that are too small to be visible and understood. Are the graphs from real data or are they make up curves and numbers? I agree that it would be useful to understand where each type of data was used for each step and what model was applied to it. But this Figure is not helping me with that. I would suggest a simplified version, for example in a diagram or schematic without images but that explains the framework, perhaps more like a workflow.

This is very valuable feedback. We have simplified Figure 1 to a simple transmission diagram that includes the data components in text form. We highlight which pieces are modeled with sub-models and which pieces are used as raw data or point estimates.

1.8 – Throughout the manuscript the authors mention "the model" to refer to the methodology or framework they develop. This is confusing because there are many models within 'the model' and the model is not actually a model per se… I suggest updating this terminology for clarity.

We have changed our terminology so that we use the term "framework" when referring to the overall strategy and "sub-model" or just "model" for a single statistical model fit to a single data type (e.g., titer curves).

1.9 – The results seem to fully hang on the definition and estimation of physiological competence. The implications should be well explained.

It seems that our language was not clear. While it is true that different choices for how to collapse the continuous titer curves (pictured in Appendix 1-Figure 1) into a single metric to compare species (whether that be AUC, max, or days above some threshold) could affect estimates of physiological competence (though we do not expect the choice to change results much given that these metrics are all very correlated), the continuous titer curves themselves are used in the next generation matrix model for calculating species' contributions to transmission and not any collapsed form of these curves. Thus, our choice of AUC has no bearing on host-to-mosquito or host-to-host transmission. To clarify this point, we have:

A) Stated this directly on lines 445-449 (for hosts) and lines 474-478 (for mosquitoes)---here we also explain that our rationale for choosing AUC was to simultaneously capture the amount and duration of titer in one metric.

B) Added an explicit statement into the Figure 2 and Figure 3 captions stating that the full titer curves, mosquito infection, and transmission curves and not the AUC summaries are used in calculating transmission; AUC is simply used to collapse a continuous curve (which are pictured in Appendix 1- Figure 1) into a metric to allow comparison among hosts.

1.10 – Why was "study" not included as random effect in the models?

A good point. We would optimally also include a random effect of study, but we had problems fitting a random effects structure with both "species" and "study" given that many studies only focused on one species. This was especially the case for mosquito transmission probability. We have added an additional caveat in the Discussion (lines 324-325) that one problem with attributing a given response to a given mosquito species with little data is a potential conflation of a mosquito species intrinsic response with a given laboratory.

1.11 – Why do a single Bayesian model for the mosquito feeding behaviour but not for the other models, and why leave out the rest of the transmission cycle? The approach is piece-meal like with multiple predictions from different GLMMs that are disconnected with one another and their summary statistics are put together at the end. Could these be put into a single Bayesian framework that allows uncertainties and data be propagated throughout the different model components?

This comment raises an important point about uncertainty propagation through the model, which we have thoroughly revised the manuscript to address. First, we would like to clarify that while we did use a piece-meal approach, we did propagate the uncertainty from each sub-model (titer curves, mosquito infection probability, mosquito transmission probability, and mosquito biting preferences), using the full density distributions for all parameters (in one of three ways---details given on lines 545-560 and in Table 1), for use in the calculations of all final results (e.g., host to mosquito and host to host transmission). We restate the point about uncertainty propagation in the methods recap sentences in the Results sections "Half-transmission cycle " and "Complete-transmission cycle " that uncertainty from sub-models was propagated.

– We decided on the piece-meal approach instead of a full Bayesian model for a few reasons. First, this approach makes it generally more user-friendly for empiricists to understand and fit a given sub-model to the data they collect. Second, this approach makes it easier to extend to other systems where given modular components can be more easily adjusted or substituted. Finally, because the sub-models relied on completely different data sets and are independent from one another, we felt like we didn't gain much from putting them all together given that we were able to extract and propagate uncertainty anyway.

1.12 – Bayesian model not defined, what was the likelihood and the priors? Provide reference for priors too.

We have expanded our explanation of the priors in the second half of the first paragraph in the Appendix 1 section “Mosquito vectors: feeding behavior”. Though we describe our rationale for the form of the priors, we do not have specific references for the exact parameters used for each. We did, however, include a citation for the general use of a Dirichlet prior for the multinomial model.

– We define the Bayesian model on lines 601-617 (including the model equation on line 608) and describe it in further detail with the two paragraphs in Appendix 1: “Mosquito vectors: feeding behavior”. We decided to keep much of the detail of this model in the Appendix in order to keep the manuscript as streamlined as possible.

1.13 – With the size of the human confidence intervals, can we make any statements about the role of human to community transmission?

We have toned down our language in the Discussion and Abstract focused on the importance of humans. However, despite wide 95\% CIs, it is still true that 32% of the density of their host-to-host transmission results lie above the upper 95% of any other species which is evidence of their importance (see Appendix 2-Figure 1). We have added this more nuanced description of the CIs and our confidence that humans are among the most competent vertebrate hosts on lines 152-155 of the Results.

1.14 – First paragraph of discussion is repetition of introduction.

We have cut this paragraph approximately in half but have kept some of it, which we like to set the tone for the Discussion.

1.15 – Table S1. Add disease associated with study.

Most of these studies are reviews and/or discussions of hypothetical frameworks and are not associated with a single disease system. We have modified the table title and caption to reflect this.

1.16 – Data does not seem available in the GitHub and the model codes need to be organised. Is the Bayesian model missing?

We appreciate you catching this omission. The missing data have been uploaded.

– The Bayesian model is present in the GitHub (mosq_bite_random.stan)

– As a road map for readers interested in looking at the code, the readme in the GitHub repository that prints to the main page states to open top_level_script.R and that script will guide the user through the other scripts. It also provides a description of all of the items in the repository. To increase ease of use we have renamed every script with a number in front to indicate the order in which files are sourced in top_level_script.R.

Reviewer #2 (Recommendations for the authors):The work presents a formidable amount of data and I particularly liked the distinction between half cycles and full cycles, and the explicit differentiation between physiological competence and exposure/behavioral processes. The finding that physiological competence may not align with epidemiological salience is an important one for vector-borne pathogens in general, and a useful lesson going forward. The paper falls short a bit on its description of the modelling process in relating them to the results, and the novelty of the finding that physiological competence does not fully explain epidemiological importance needs to be clarified, but overall this is a solid and useful contribution. The following points should be addressed

We appreciate this thoughtful and positive feedback.

2.1 If the paper is going to be in results-first format, the results need a bit more description of the models to facilitate understanding. I found myself wondering how exactly these results were being produced, and needing to flit back and forth from the results to the methods, which isn't ideal. Including the right level of detail (i.e. without recapitulating the methods) is tricky in a results-first Results section, but I think this paper could definitely do with including more information. A few sentences at the start of each section giving a bit of background on the model formulation and how the answers were produced would do the job.

Thank you for this valuable suggestion. We have opted for one to two recap/highlight sentences at the start of each Results section to give the reader an overview of the most important methods used in each Results sub-section. We have also added a ‘Box 1” in the Introduction (and tie it to the revamped Figure 1) which also provides some expanded detail which will hopefully help with the Results-first format.

2.2 The models are doing a very heavy lift here (particularly the NGM's), and I appreciate the authors' decision to include model limitations fairly prominently in their discussion. I have no personal experience with models like these, but they appear to have been conducted well.

Thanks.

2.3 Although the authors' reference to host species' ecology as an important determinant of their epidemiological role is useful, the discussion particularly could do with zeroing in more precisely on what aspects of the hosts' ecology are driving the disparity between physiological competence and epidemiological importance (i.e., behaviors, vector biting preferences, and population dynamics). The findings in the paper are definitely novel, but without clarifying what might be going on to drive this disconnect there's a risk of reinventing the wheel of behavioral competence a bit. These processes are all decomposed in Cynthia Downs and colleagues' 2019 competence paper in Trends in Parasitology (for example), and I think the paper should be a bit more explicit in the introduction and discussion that there are well-appreciated reasons that we might not expect physiology to paint the whole picture of competence. For example, it is relatively unsurprising that very rare hosts are epidemiologically unimportant in the system irrespective of their physiological competence. Is it possible that humans' role in these dynamics could be reduced to "there are lots of humans, so they end up being important"? The fact that physiological competence is only one component is in the introduction (lines 54-58), but a more detailed outline of the traits (behavioral and demographic) that could override it is necessary. This will then set the stage for the findings of the models.

This is a great point. First, we have rephrased and expanded paragraph 3 of the Discussion (lines 237-260) to open with a statement that others have thought about changes between physiological and ecological ability citing Downs et al., 2019 and Simpson et al., 2012. In the next sentence we cite 4 papers that have found strong discrepancies between physiological and ecological importance. Then, near the end of the paragraph we talk somewhat broadly that one strength of the three step approach is that it can help reveal what ecological factor matters for a host or vector species' changing importance. We have also expanded our Discussion paragraph 4 (lines 261-277) as a continuation of this topic to describe with more nuance what is driving human and bird species transmission importance in RRV. As you say, humans are indeed abundant so increase in importance. However, this is not the full story as they are a moderately preferred sources for blood meals by moderately competent mosquitoes and also have a stronger physiological response than they have previously been given credit for. We contrast this scenario with birds, which are much less abundant but have a very high feeding preference for one of the most competent vectors.

– Finally, we rephrased the second paragraph of the Introduction to clarify that our framework is not presenting an entirely new strategy for thinking about competence and vector-borne disease transmission (e.g., lines 37-39). Rather, that it is a data-driven synthesis and analysis method that streamlines previously proposed ideas.

[Editors' note: further revisions were suggested prior to acceptance, as described below.]

1 – Figure 1 is much more informative now but perhaps could be expanded to also provide an indication of the type of model used in each parameter to show the workflow.

We now note in the bottom right of the figure the type of model used for each of the black-boxed terms (the terms fit with statistical models).

2 – On previous comment 1.1. I appreciate the revisions and the more honest presentation of the results. All much clearer. I also welcome the explanation of propagation of uncertainty, this is an important feature that I have missed. One of my key points here though was that regardless of how uncertainty is estimated, a formal quantitative way of comparing the densities estimated (such as z-test, Wilcoxon or Kolmogorov tests?) could strengthen the outcomes.

We respectfully disagree that a statistical test is necessary to strengthen the outcomes. Given that we propagated uncertainty and calculated 95% CI on the resulting densities we prefer relying on the simple heuristic of overlapping CI, which we find easy to interpret and visualize.

3 – On previous comment 1.11. I don't agree with the reasons mentioned for not using a single Bayesian model instead of the piece-meal approach – 2/3 are considered advantages of Bayesian modelling and the less-friendly approach is debatable (indeed the framework proposed in this manuscript does not seem very user-friendly for empiricists either). That said, although I would approach it differently, I cannot say it is incorrect – though I raise here the concern about how Bayesian and ML-type outcomes are combined in the NGM. The outcomes of a Bayesian model are fundamentally different from those so-called frequentist approaches. All your parameters are 'frequentist' apart from feeding behaviour. Are these being incorporated in a comparable way?

We appreciate your concern about combining parameter estimates in a piece-meal approach. We describe our strategy for estimating probability densities for each parameter fit using Frequentist methods on lines 548-555. By constructing these probability density functions (using the variance-covariance of the estimates and assuming multivariate normality) we were able to combine the Frequentist and Bayesian estimates by sampling both the Frequentist probability density functions of the parameter estimates and the Bayesian posteriors. Finally, while we agree that fitting the whole model using our approach may not be more empiricist-friendly than fitting a Bayesian model, using a piece-meal model does allow an empiricist to use a single code chunk to fit a single model which is not possible with one giant Bayesian model.

4 – Titre profiles. I have two standing concerns with these. Not sure if problems or if they just need to be explained and discussed.

Thank you for raising these points. We gave considerable thought on how to combine experimental data from different studies that reported only summary statistics. We believe our method appropriately captures the variation among the individuals of a given species and the number of individuals exposed (e.g., more variation and a smaller sample size produces wider uncertainty bands as shown in Appendix Figure 1). We clarify further below.

4.1. Titration values tend to be highly variable within species, let alone between species. Given the small differences in the results, how confident can we be in that the results are meaningful at the ecological scale used?

It is true that individuals vary strongly in their response to infection. However, this variation is captured in our statistical model (see Appendix 1 Figure 1; note that this variation is very large given that the y-axis is log10 titer). Given that the uncertainty in host responses is propagated through the model (to the probability that a mosquito will become infected given a bite, which then affects the number of second generation hosts they will infect), any clear results (e.g., that rats have the highest physiological competence) are clear despite this uncertainty. We also note that including the proportion of individuals that generated a viremic response is, by itself, an advancement from all previous models for RRV which only included the viremic response of a single individual and did not consider any variation within species.

– It is also true that the variation among individuals within and between species may cause the dynamics of a local emerging epidemic to not be predicted well with a model using the mean response of all species. However, this is not the primary goal of our research. Given that our primary aim was to estimate the relative importance of each species on a larger scale on average, we believe our findings are meaningful at the scale we chose. Finally, we are careful not to overstate confidence (e.g., the caveats in the Discussion) given our various simplifications/assumptions regarding ecological scale (e.g., homogeneous mixing and species abundances from surveys over a small area perfectly representing their abundances over Brisbane at large).

4.2. The analysis are based on peak titre but considering the different profiles this could be problematic. E.g. in the example shown (Appendix 2 Figure 7: A) where one species has a marked high peak and the other has a constant titre lower than the peak of the other species. How can we link this to an infection/transmission probability? Especially the short time frame of the experiments.

We agree very strongly with this point, and it supports the approach that we did use, which was not based on peak titer. We use the full time-dependent titer curve to calculate mosquito infection probability by translating the actual time-dependent titer values (y-axis on Appendix 1 Figure 1) to mosquito infection probability (x-axis in Appendix 1 Figure 3). This is stated in the Equation.1 term p_j_ | θ_id_ which indicates that mosquito infection probability (p_j_) is a function of the titer in the host species *i* that the mosquito bites on day d of that host’s infections period (θ_id_) (described on lines 497 and 498). We have now modified lines 437-438 to more explicitly state that we modeled “continuous functions of titer over time”. Further, we have now adjusted the caption of Figure 2 to read “…time-dependent [previously “raw”] titer profiles (pictured in Appendix 1 Figure 1) are used in half-cycle and complete-cycle transmission.” As you note, a high but short-lived viremia may not lead to more infections in mosquitoes than a low but long-lived viremia—by using the full titer curve and translating the values into mosquito infection probabilities we capture this possibility. The only place we discuss peak viremia is in the Methods section on the titer curves (lines 425-450) where we describe in detail how we translated peak and duration as reported in empirical studies into continuous titer curves. On lines 436-439 we explicitly state: “For non-human species, only means and standard deviations for peak titer and duration of detectable titer were reported. We transformed these summary measures into continuous titer profiles (continuous functions of titer over time that are needed to quantify mosquito infection probability) by modeling titer profiles as quadratic functions of time since infection, based on observed patterns in the data.” Finally, to quantify host physiological competence we calculate the area under the viremia curve rather than peak viremia alone (as explained on lines 91-96 and 445-450).

5 – Appendix 1 Figure 4: Still don't understand how these curves are generated. I appreciate that a baseline curve is being generated from all the data combined (all studies and species). But then how can you differentiate among species. A species random effect is added but, on most occasions, there is 1 transmission value for 1 time point for a single species. Did the model pass validation diagnostics? There is also some confusion with the use of dose: the legend first mentions that dose was used as a fixed effect and then a few lines down says it couldn't be used as fixed effect. This also raises issue about the meaning of the transmission probability, especially post ~15 days as all is driven by a single species (no data for all others).

First, thanks for catching the discrepancy with dose; the text has been corrected. As we correctly state in the manuscript’s Methods, dose was not used as a fixed effect in the transmission probability regression model.

– Second, while it is true that few species have data points after day 15, this has relatively little effect on the logistic regression because many high probability values (i.e., a high proportion of mosquitoes were found to be able to transmit) were recorded prior to day 15. Given that the logistic regression assumes a response will eventually reach the asymptotic probability of 1 (though we note that transmission probability is scaled by survival probability so a transmission probability of 1 is never realized by a mosquito), a number of high probability values prior to day 15 strongly constrains the location of the inflection point and thus the probabilities after day 15. That is, since a number of species approach a high transmission probability prior to day 15 (e.g., *Ae. camptorhynchus, Ae. notoscriptus, Ae. aegypti, Ae. vigilax, Cq. linealis*), the model is able to estimate (albeit with relatively large uncertainty) the inflection point of the logistic curve and thus the probability at higher values (> 15) of the covariate of “day”.

– Finally, to the point about generating the curves, the model is fit with lme4 and passed validation diagnostics. To generate the curves for each species we:

A) extracted the fixed effects estimates and variance–covariance matrix for the “grand mean” intercept and slope over day.

B) extracted the species-level deviates (i.e., the conditional modes of the random effect). A fitted lme4 model object in R stores the estimates for the conditional modes (here how much each species’ intercept and slope over time deviate from the fixed effects (grand mean) estimates) and the uncertainty on those estimates (in their own variance-covariance matrix). To say it in another way, these are the samples from the zero-centered multivariate Normal distribution that describes the variance in the random effects (intercept and slope over time).

We state that we use these “conditional modes and conditional covariances of the random effect” in two places: in the Appendix 1 text (at the end of the section “Mosquito vectors: infection and transmission probability”) and in the Appendix 1 Figure 4 caption. In both spots we direct the reader to the code hosted on GitHub for more detail. The R code contains commenting about steps A and B and also step C detailed below. We also state, albeit in abbreviated form, that we construct density distributions using the variance covariance matrix of the estimated coefficients on lines 550-554.

C) In brief, a mixed regression model—here the transmission probability of mosquito species *i* on day *j*—can be written: logit(y_ij_) = α + a_i_ + (β + b_i_)*x_j_ (where α and β are the fixed effects intercept and slope estimates, a_i_ and b_i_ are the estimated species-level intercept and slope deviates, and x_j_ is day *j*). To obtain uncertainty on yij we combine 1000 samples for the intercept and slope estimates (α and β) (using the covariance matrix for these terms) with 1000 samples from the conditional modes (a_i_ and b_i_) (using the covariance matrix for the conditional modes of the random effect).

The grand mean intercept and slope over day are estimated reasonably well in the model, while the estimate for individual species (the ai and bi terms) are sometimes estimated with little uncertainty and sometimes with a lot of uncertainty (e.g., see *Ae. alternans*). Specifically, in the case that the data point for a given species resides far from the overall estimated mean response (e.g., *Ae. alternans* has a lower than average transmission probability on day 10), the uncertainty gets very large, which is sensible —the model essentially is saying: it doesn’t seem like this species is responding similar to the average, but it is very unclear what its response is. A curve can still be drawn, but as can be seen in the *Ae. alternans* panel, the 95% intervals at day 20 span almost the whole range of probabilities (nearly 0-1).

6 – Appendix 1 Table 1: swap word 'reservoir' for 'host' to be consistent with main text.

Done

7 – Figure 2 add sample sizes from each species.

These are modeled outputs and not raw data associated with specific sample sizes. For example, Figure 2B and 2C are showing the total number of mosquitoes or hosts a single viremic host can go on to infect, while Figure 2A represents the viremic potential for a single infected host compared to other infected hosts. We have added the term “estimated” to the description of each panel to help clarify this point. The sample sizes for the raw data used in all of the statistical models that contribute to this figure are located in the Appendix Tables; we do not feel that they should be re-reported here.

– L98-100. cattle, horses and flying foxes seem all pretty similar to me

Indeed. Thanks for catching this mistake in our language. We have updated this sentence to read that birds had a stronger viremic response that flying foxes, horses, and cattle.